# Low-energy electronic structure in the unconventional charge-ordered state of ScV$_6$Sn$_6$

Asish K. Kundu[1,10], Xiong Huang [2,10], Eric Seewald[2], Ethan Ritz [3], Santanu Pakhira [4,9], Shuai Zhang [2], Dihao Sun[2], Simon Turkel [2], Sara Shabani[2], Turgut Yilmaz [5], Elio Vescovo[5], Cory R. Dean [2], David C. Johnston[4,6], Tonica Valla [7], Turan Birol [3], Dmitri N. Basov [2], Rafael M. Fernandes [8] & Abhay N. Pasupathy [1,2] ✉

Kagome vanadates $A$V$_3$Sb$_5$ display unusual low-temperature electronic properties including charge density waves (CDW), whose microscopic origin remains unsettled. Recently, CDW order has been discovered in a new material ScV$_6$Sn$_6$, providing an opportunity to explore whether the onset of CDW leads to unusual electronic properties. Here, we study this question using angle-resolved photoemission spectroscopy (ARPES) and scanning tunneling microscopy (STM). The ARPES measurements show minimal changes to the electronic structure after the onset of CDW. However, STM quasiparticle interference (QPI) measurements show strong dispersing features related to the CDW ordering vectors. A plausible explanation is the presence of a strong momentum-dependent scattering potential peaked at the CDW wavevector, associated with the existence of competing CDW instabilities. Our STM results further indicate that the bands most affected by the CDW are near vHS, analogous to the case of $A$V$_3$Sb$_5$ despite very different CDW wavevectors.

The metallic kagome vanadates have recently gained prominence due to the variety of emergent electronic behaviors that they exhibit[1]. These include experimental reports of charge density waves (CDWs)[2–10], superconductivity[9,11–13], pair density waves[14], rotational symmetry breaking[15–18], and time-reversal symmetry broken states[19–25], particularly in the family of compounds $A$V$_3$Sb$_5$, with $A$ = K, Rb, Cs. These materials feature a complex electronic structure with several vanadium orbitals contributing to states near the Fermi level ($E_F$), resulting in flat bands, Dirac cones, and multiple van Hove singularities (vHSs) that are reminiscent of the nearest-neighbor tight-binding

model on the kagome lattice[26–30]. Several ARPES measurements together with theoretical studies have now been performed on the $A$V$_3$Sb$_5$ materials, resulting in a convincing description of their normal-state band structure[11,31–33]. While the precise microscopic origin of the CDW instability remains under investigation, it is evident that electron-phonon interactions play a crucial role, as attested to by the softening of the zone-corner phonon mode found in first-principles calculations and confirmed by phonon-spectroscopy measurements[5,7,10,34–37]. Given the presence of unusual electronic phenomena in the low-temperature phase of the kagome vanadates, it is paramount to elucidate to what

---

[1]Condensed Matter Physics and Materials Science Division, Brookhaven National Laboratory, Upton, NY 11973, USA. [2]Department of Physics, Columbia University, New York, NY 10027, USA. [3]Department of Chemical Engineering and Materials Science, University of Minnesota, Minneapolis, MN 55455, USA. [4]Ames National Laboratory, Iowa State University, Ames, Iowa 50011, USA. [5]National Synchrotron Light Source II, Brookhaven National Laboratory, Upton, NY 11973, USA. [6]Department of Physics and Astronomy, Iowa State University, Ames, Iowa 50011, USA. [7]Donostia International Physics Center (DIPC), 20018 Donostia-San Sebastián, Spain. [8]School of Physics and Astronomy, University of Minnesota, Minneapolis, MN 55455, USA. [9]Present address: Department of Physics, Maulana Azad National Institute of Technology, Bhopal 462003, India. [10]These authors contributed equally: Asish K. Kundu, Xiong Huang. ✉e-mail: apn2108@columbia.edu

extent the CDW order impacts the electronic spectrum deep in the ordered state. In the $AV_3Sb_5$ compounds, a prominent reconstruction of the band structure is clearly observed by ARPES, revealing a mechanism by which the interactions driving the CDW instability may impact superconductivity and other types of electronic order[8,38-40].

In the other family of kagome metals, $RV_6Sn_6$, with $R$ = Sc, Gd, Y, Ho, only the Sc compound has been reported to display CDW order[41]. While $ScV_6Sn_6$ shows a CDW order at a similar ordering temperature (~ 92 K) as the $AV_3Sb_5$ compounds, its wave-vector is $\tilde{K} = (1/3, 1/3, 1/3)$[41], in contrast to the $L = (1/2, 1/2, 1/2)$ wave-vector of the latter. The normal-state band structure of the $AV_6Sn_6$ family has broad similarity with that of the $AV_3Sb_5$ family[42,43], with Dirac cones and multiple vHSs near $E_F$. Unlike the $AV_3Sb_5$ compounds[32,33], however, there is no clear relationship between the CDW wave-vector $\tilde{K}$ and the vHSs located at $M = (1/2, 1/2, 0)$. Moreover, in contrast to $AV_3Sb_5$, where DFT calculations predict the experimentally-observed CDW wave-vector, in $ScV_6Sn_6$ the leading CDW instability predicted by DFT has a different out-of-plane wave-vector component[37,44,45], raising questions about the nature of the CDW phase. Little is currently known experimentally about the electronic structure or interactions in the CDW phase of the $ScV_6Sn_6$ compound. Therefore, elucidating the differences and similarities between the electronic spectra inside the CDW-ordered phases of $ScV_6Sn_6$ and $AV_3Sb_5$ is crucial to understand the origin of the different low-energy electronic properties of these two families of kagome metals. In this work, we address this question with nano-optical spectroscopy, angle-resolved photoemission spectroscopy (ARPES), and scanning tunneling microscopy (STM) measurements, combined with theoretical modeling.

## Results

The lattice structure of $ScV_6Sn_6$ above the CDW transition temperature $T_{CDW}$ ~ 92 K is shown in Fig. 1a, and is described by the space group P6/mmm. A single unit-cell contains two V layers sandwiched by $ScSn_2$ and Sn layers. Viewed from the top, the V atoms form kagome layers with

an in-plane lattice constant of $a = 0.55$ nm. As reported by diffraction experiments[41], below $T_{CDW}$, the lattice exhibits a ($\sqrt{3} \times \sqrt{3}$) lattice distortion in the $a - b$ plane. This CDW transition causes a sharp change in the infrared nano-optical response of the material, as revealed by the nano infrared (nano-IR) scattering amplitude plotted in Fig. 1b, which is also consistent with a recent optical spectroscopy study on the same material[46]. The electronic response to the CDW lattice distortions can be directly visualized with STM. Shown in Fig. 1c is an STM topograph of the vanadium layer at $T = 6.7$ K, deep in the CDW phase. The real-space image shown in Fig. 1c (also the line profile over the surface in Fig. 1e) and its Fourier transforms in Fig. 1d clearly show a CDW peak located at (1/3, 1/3) with respect to the Bragg peak (see Supplementary Fig. 1 for evidence of a three-dimensional charge order). STM topographic images of the $ScSn_2$ surface also show the presence of charge order with the same ordering vector (Supplementary Fig. 2). Tunneling spectra for both surfaces (Fig. 1f) show similar spectral features with a prominent peak in the electronic density of states (DOS) at around − 65 meV, which is related to the vHS as evident from our ARPES results (see following discussion).

To understand the momentum-space electronic structure, ARPES measurements were performed both above and below the CDW transition temperature. Figure 2a shows the Fermi surface (FS) measured on the $ScSn_2$-terminated surface of $ScV_6Sn_6$ above $T_{CDW}$. Most of the FS features are captured well by our DFT-calculated FS (bulk) (Fig. 2g) provided that a small upward shift of 20 meV is applied, possibly due to small unintentional doping of the crystal during growth or due to surface effects. Additional features seen in the experiment in the second Brillouin zone originate from the surface states (SS) of the $ScSn_2$-terminated layer[37]. At the $\bar{K}$ point, a small circular and two nearly-triangular Fermi pockets are observed. A small rectangle-shaped pocket at $\bar{M}$ can be identified connecting the adjacent triangular Fermi pockets. Band dispersions along cut #1 (Fig. 2c) and cut #2 (Fig. 2e) show that the outer and inner triangular FSs are formed by the hole-like ($\alpha$) and electron-like ($\beta$) bands, respectively. Dirac-like band

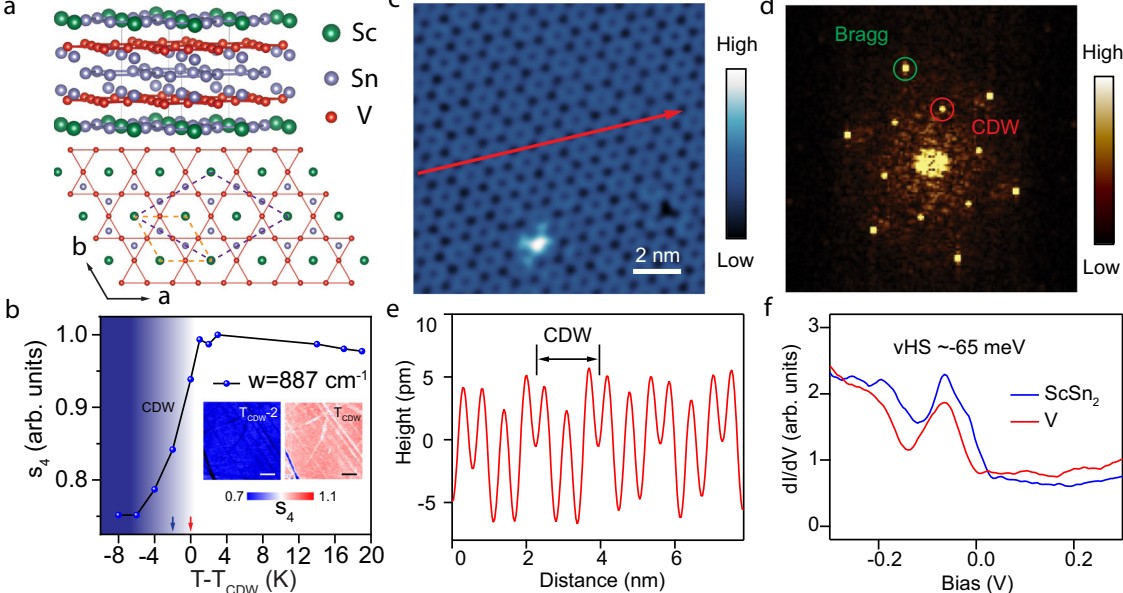

**Fig. 1 | Lattice structure and CDW in $ScV_6Sn_6$. a** Crystal structure of $ScV_6Sn_6$ (above the CDW transition). The V kagome layer is sandwiched by $ScSn_2$ and Sn layers. Gray lines outline the unit cell. The view from the top shows the V-based kagome layer. Orange and purple dashed diamonds indicate the in-plane unit cell above and below the CDW transition temperature, respectively. **b** CDW phase transition revealed by nano-infrared (nano-IR) images. The normalized nano-IR scattering amplitude $s_4$ plotted as a function of temperature. Inset: images of $s_4$ at $T_{CDW}$ and 2 K below it, respectively. Scale bar: 0.5 μm. **c**, STM topography (0.2 V, 100 pA) of the V-terminated surface in the CDW phase. **d** The Fourier transform of **c** with the CDW and the atomic Bragg peaks marked by red and green circles, respectively. **e** Line profile extracted along the red arrow in **c**, showing the lattice reconstruction due to the CDW. **f** $dI/dV$ tunneling spectra on different surface terminations, as indicated. The peaks at the density of states just below $E_F$ correspond to the vHSs seen in ARPES.

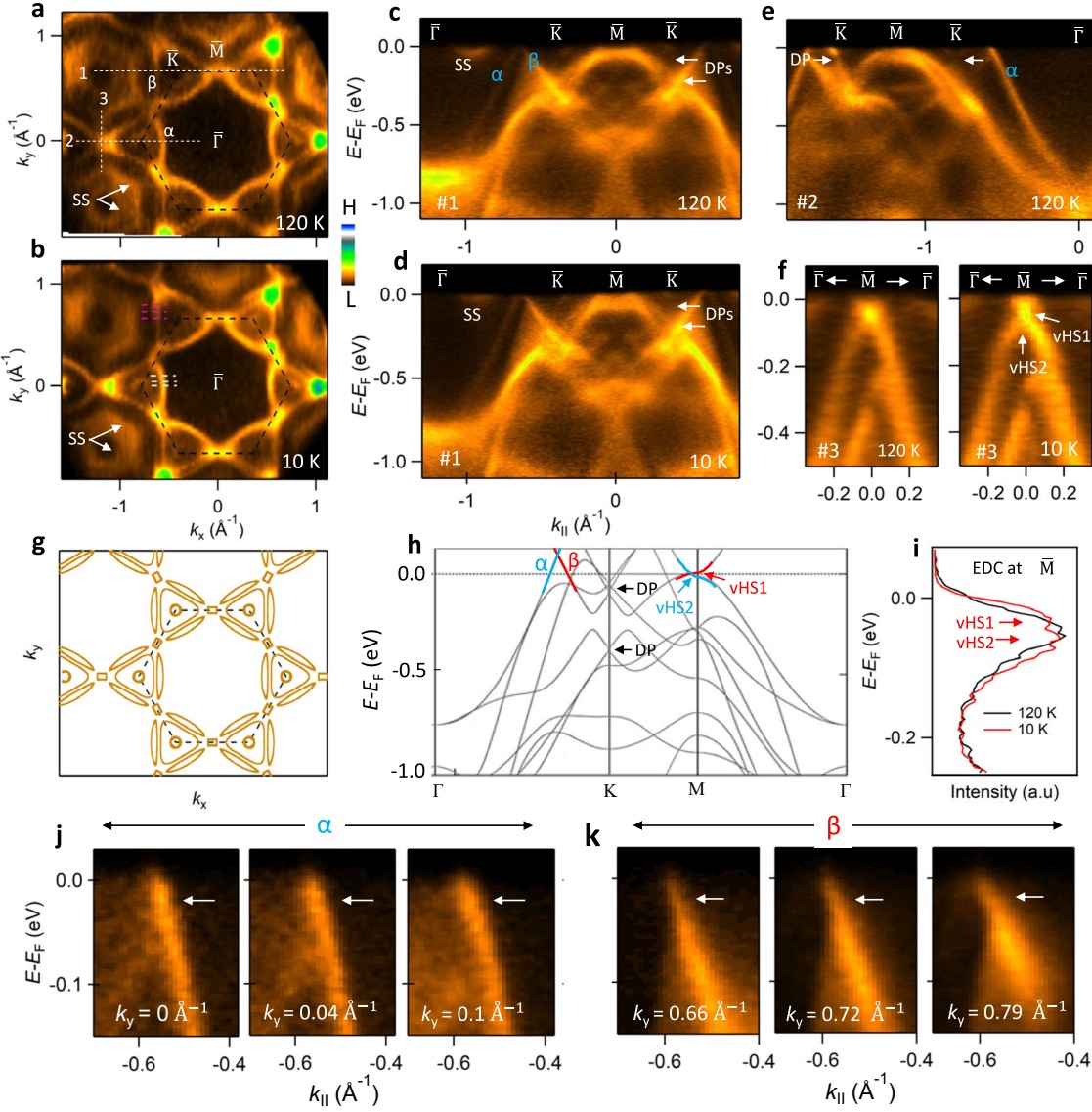

**Fig. 2 | Mapping the electronic band structure of ScV₆Sn₆.** **a** and **b** Fermi surface map above and below the CDW phase transition temperature (~92 K), respectively, obtained from ARPES measurements using $h\nu = 110$ eV ($k_z \sim 0$). **c** and **d**, Band dispersion along the direction shown by line 1 in **a** and **b**, respectively. **e** Band dispersion along line 2 at 120 K. Bands marked by $\alpha$ and $\beta$ form two nearly triangular pockets at $\bar{K}$ as indicated in **a**. Arrows in **c**–**e** indicate the Dirac points (DPs). A surface state (SS) originating from the ScSn₂ terminated layer[37] is indicated in **a**–**d**. **f** Band dispersion along line 3, above (left) and below (right) the CDW transition. The vHSs (vHS1 and vHS2) are indicated by arrows in **f**. **g** DFT-computed constant-energy contour at $E = 20$ meV and $k_z = 0$ plane for the bulk non-CDW structure. **h**, Theoretical electronic structure along the $\Gamma$-K-M-$\Gamma$ path. The bands forming the vHS at the M point are highlighted. Black arrows indicate the Dirac points at the K point. **i**, EDCs at $\bar{M}$ for 10 K and 120 K. **j** and **k**, ARPES spectra along various line cuts through the $\alpha$ ($\beta$) band, respectively, as indicated by a series of white- and pink-dashed lines in **b**. Arrows indicate the location of a kink.

crossings are also observed at $\bar{K}$ at energies of −70 meV and −260 meV (labeled by DP in Fig. 2c–e). In the FS maps (Fig. 2a, b), high spectral intensity is observed around the $\bar{M}$ points of the Brillouin zone (BZ) due to the close proximity of the vHS to the Fermi energy ($E_F$). Indeed, two vHSs are observed at $\bar{M}$ at energies of −25 and −60 meV (Fig. 2f), as expected from our DFT calculations (Fig. 2h), and in agreement with the DOS peak seen in the tunneling spectra of Fig. 1f. The vanadium-terminated surface also shows very similar ARPES spectra as the ScSn₂-terminated layer (Supplementary Fig. 3). Furthermore, the electronic states forming the FS show moderate $k_z$ dispersion at $E_F$ (Supplementary Fig. 4).

The ARPES Fermi surface and band dispersion along cut #1 measured inside the CDW state are shown in Fig. 2b and d, respectively. In general, we find nearly identical electronic structures inside and outside the CDW states, with the only minor difference being the energy

position of the vHS (Fig. 2i and Supplementary Fig. 5). In contrast, a much larger energy shift (~90 meV) of the vHS has been seen in the $A$V₃Sb₅ materials[32,47]. Importantly, the band folding and energy gaps at the Fermi surface seen by ARPES in the $A$V₃Sb₅ materials[32,38–40] are not seen here in ScV₆Sn₆, consistent with a recent optical spectroscopy study[46]. In Fig. 2j, k, we show the electronic dispersions of the $\alpha$ and $\beta$ bands for various $k_y$ momenta. A kink-like feature appears around −25 meV (indicated by arrows) and becomes more prominent while approaching the $\bar{M}$ point (higher $k_y$ in this figure). Such kinks are typically associated with momentum-dependent electron-phonon coupling, although additional analyses are required to establish their origin in ScV₆Sn₆[36,48,49]. The lack of a visible band reconstruction indicates that the coupling between the CDW and the electronic structure is weak, which suggests that electronic interactions do not play a prominent role in promoting the CDW. This is in line with recent

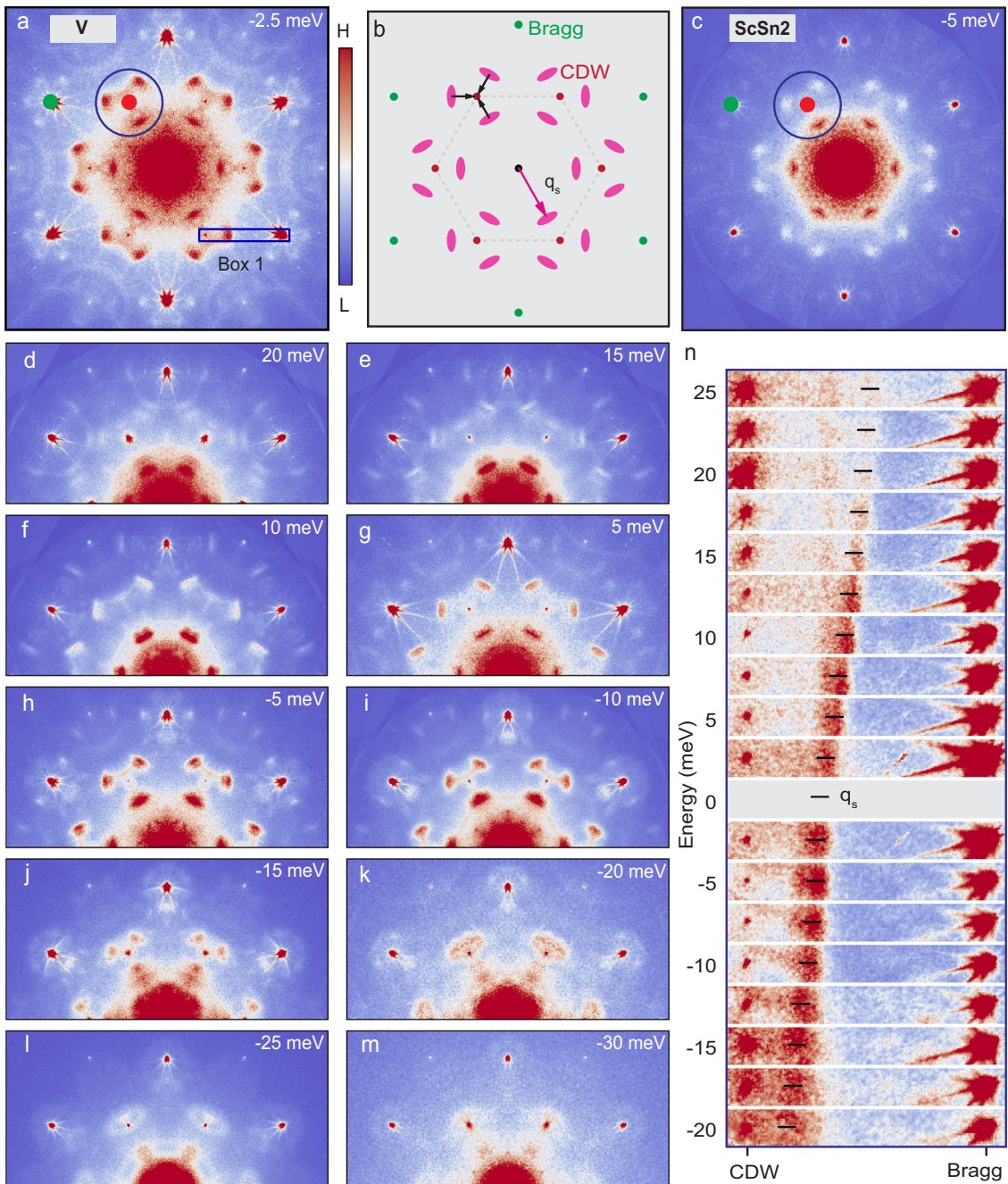

**Fig. 3 | Quasi-particle interference imaging of the V kagome and ScSn₂ surfaces.**
**a**, **c** FT-STS image near the Fermi level for V kagome and ScSn₂ surfaces, respectively. The real-space image was acquired over a 150 nm field of view. Green and red dots mark the positions of Bragg and CDW peaks, respectively. The blue circle highlights the prominent QPI spots surrounding the CDW peaks. **b** Schematic drawing of the QPI features shown in **a**. The most prominent QPI features (at a momentum $\mathbf{q}_s$ indicated by the pink arrow) are 3-fold symmetric with respect to the CDW peaks. Other features surrounding the Bragg peaks are discussed in the Supplementary Fig. 6. **d**–**m** QPI images taken at different energies as indicated in each panel for V kagome surface. **n** Zoomed-in view of the region in Box 1 of panel

**a** at different energies. It is clearly seen that $\mathbf{q}_s$ (labeled by the pink arrow in **b**) is highly dispersive and merges into the static CDW peaks at $E \sim -30$ meV in (**n**). The color scale in each image has been adjusted independently for visualization. STM setup ($V_{sample}$, $I_{set}$, $V_{exc}$) conditions for each panel: **a**, (−2.5 mV, 110 pA, 1 mV); **c**, (-5 mV, 120 pA, 2 mV); **d**, (20 mV, 125 pA, 2 mV); **e**, (15 mV, 120 pA, 2 mV); **f**, (10 mV, 110 pA, 2 mV); **g**, (5 mV, 110 pA, 1 mV); **h**, (−5 mV, 115 pA, 1 mV); **i**, (−10 mV, 120 pA, 2 mV); **j**, (−15 mV, 120 pA, 2 mV); **k**, (−20 mV, 130 pA, 1 mV); **l**, (−25 mV, 145 pA, 2.5 mV); **m**, (−30 mV, 140 pA, 1 mV). For zoom in data shown at other energies in **n**, STM setup conditions are listed in the Supplementary Fig. 6.

theoretical calculations and Raman spectroscopy results arguing that the CDW transition in ScV₆Sn₆ is a lattice-driven instability[37,44].

From the above discussion, it is evident that the effect of the CDW on the electronic structure is subtle, if present at all, and not easily discerned in ARPES measurements[37,50–52]. To further investigate this issue, we turn to Fourier-transform scanning tunneling spectroscopy

(FT-STS) measurements, where quasiparticle interference can be used as a sensitive probe of the electronic structure[2,16,53–56]. Shown in Fig. 3 is a set of FT-STS maps obtained on the V-layer surface over a range of energies near the Fermi level (more data can be found in Supplementary Fig. 6). Importantly, similar behavior is observed on the ScSn₂ surface termination, as shown in Fig. 3c and Supplementary Fig. 7,

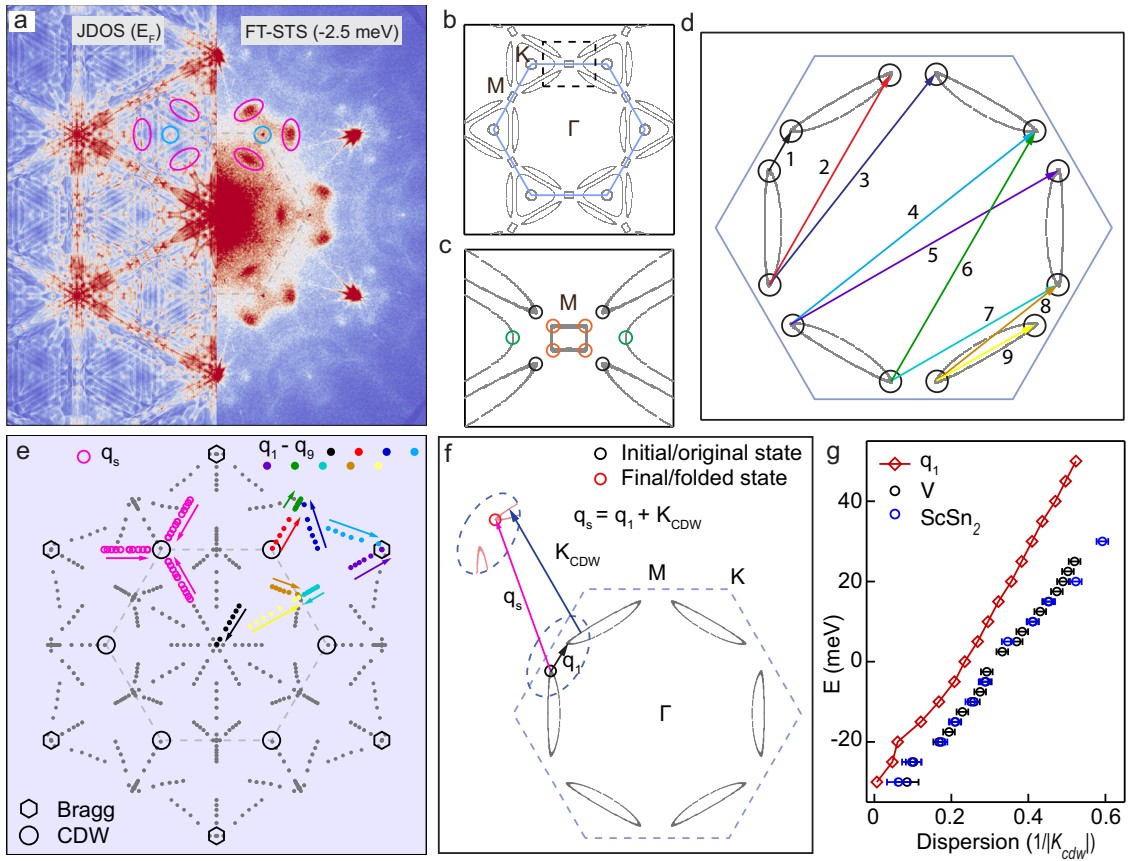

**Fig. 4 | Band folding to explain the observed QPI features. a** Direct comparison between the normal state JDOS from DFT calculations (left) and QPI (right) at the Fermi level with the QPI features and CDW peak positions marked. The prominent spots seen in the experimental QPI are not observed in the normal state JDOS (pink ellipses). **b** Fermi surface calculated from DFT. **c**, Zoomed-in image [dashed box of (**b**)] showing all the possible hot-spots of the band structure as highlighted by the colored circles. **d**, Scattering wave-vectors indicated by the 9 independent colored arrows between the hot-spots (black circles). **e** All experimental scattering wave-vectors (gray dots) within the energy window of [30, −30] meV, superimposed on the 9 independent wave-vectors connecting the hot-spots (colored dots). The arrows indicate the expected energy dispersion from 30 to −30 meV. The experimentally observed QPI dispersion is shown by the pink circles and pink arrows over the same energy range. **f** Illustration of band folding of states at the hot-spots by one CDW wave-vector (blue arrow). $q_s$ corresponds to a scattering process with an initial state at the hot-spot of the normal state and a final state at the hot-spot arising from band folding, as indicated by the pink arrow. **g**, Comparison of the dispersion for the experimental QPI ($q_s$ from two different surface terminations with error bars defined) and hot-spot scattering wave-vector ($q_1$) from theory.

indicating that the QPI is not sensitive to the specific details of the surface. The FTs have been symmetrized with respect to the point group symmetries of the crystal to enhance the signal-to-noise ratio (see Supplementary Fig. 8 for raw real and Fourier-space data). A typical FT-STS map (Fig. 3a at energy $E = -2.5$ meV) shows an atomic Bragg peaks (green dot), the CDW peaks (red dot), and several other features, which disperse strongly with energy, allowing us to associate them with quasiparticle interference (QPI). The most prominent of these features are a set of three spots that are 3-fold symmetric about the CDW peak. These spots, located at momenta $q_s$, are encircled by the blue circle in Fig. 3a and schematically shown in Fig. 3b together with the Bragg and CDW peaks. Figure 3d−m illustrate the dispersion of these spots as a function of energy, within the energy range of $\pm 30$ meV around the Fermi level where they are visible. In Fig. 3n, zoomed-in images display the dispersion of these QPI features (box 1 in Fig. 3a). The QPI spots disperse towards the CDW wave vector as the energy is decreased from +30 meV, finally merging with the CDW peak at $E \approx -30$ meV.

Within the Born approximation, the amplitude for scattering from a point-like scatterer is proportional to the density of initial and final states. In this case, a simple way to understand the FT-STS images is by making a comparison with the joint density of states (JDOS), or equivalently, the autocorrelation of the spectral function $A(\mathbf{k}, E)$. While the formal definition of the QPI is different from the JDOS (See

Section 1.3 in the Supplementary Information for discussion on this), we expect that when the spectral intensity of the folded bands is weak, both the JDOS and QPI are small for scattering that involves the folded bands. Shown in Fig. 4a is such a comparison between the normal-state JDOS from DFT and the experimental FT-STS near the Fermi level. Since the ARPES data show no discernible difference between the electronic spectra above and below the CDW transition, we use the DFT band structure of the normal state. We see that the corresponding JDOS completely fails to reproduce the prominent dispersing spots at $q_s$ seen experimentally (pink ellipses in Fig. 4a; comparisons at other energies are shown in Supplementary Fig. 9). We must therefore look for explanations beyond the simple model for QPI presented above.

To gain further insight into the observed QPI, we notice that it takes the form of spots in the FT. Since scattering wave-vectors connect initial and final states, the presence of discrete spots in the QPI implies that the scattering is dominated by "hot-spots" in the band structure, i.e., regions of high density of states $\nabla_{\mathbf{k}}E(\mathbf{k})^{-1}$. This idea has been extensively utilized to study the band structures extracted from FT-STS studies of high-$T_c$ superconducting cuprates[57]. In the Fermi surface of $ScV_6Sn_6$, shown in 4b, a number of such hot-spots exist near the **M**-point vHS (zoomed in Fig. 4c). As a result, one expects that the dominant QPI scattering wave-vectors are those that connect any two of these hot-spots to each other. From a consideration of the magnitudes and the energy dispersions of all these possible scattering

vectors (see Supplementary Fig. 10), we concentrate our discussion below on a subset of the hot spots, marked by black circles in Fig. 4c, d. Our simplified model consists of replacing the normal-state DFT Fermi surface by these 12 hot-spots. Scattering between the hot-spots gives rise to 9 independent wave-vectors as labeled with colored arrows in Fig. 4d. All other scatterings between these hot-spots are related to these 9 vectors by crystal symmetries. We carefully examine the dispersion of these scattering wave-vectors and plot them in Fig. 4e. The positions of the prominent experimentally observed QPI at $\mathbf{q}_s$ discussed above are displayed around one of the CDW Bragg peaks as pink open circles (their dispersion labeled by pink arrows). None of the normal-state hot-spot scatterings match the strongest features seen in the QPI - a result that is already anticipated by the fact that the normal-state JDOS cannot reproduce these experimental features. Weak signatures of these normal-state hot-spot scattering are indeed seen around the atomic Bragg peaks, as shown in Supplementary Fig. 6.

Although the normal-state hot-spot scattering vectors fail to match the dispersion of the spots (pink) seen inside the CDW phase at $\mathbf{q}_s$, we notice that when these scattering vectors are shifted by the CDW wave-vector $\mathbf{K}_{CDW}$, as illustrated in Fig. 4f, they fall on top of the experimentally observed features. Mathematically, this means that $\mathbf{q}_s - \mathbf{K}_{CDW}$ matches one of the scattering vectors involving the hot-spots of the normal-state bandstructure. Equivalently, this can be interpreted as a scattering vector connecting a hot-spot of the normal-state bandstructure with another hot-spot of the folded bandstructure (i.e. the normal-state bands shifted by the CDW wave-vector as shown in Fig. 4f). The dispersion of the simulated QPI from the hot-spot model ($\mathbf{q}_1$) matches quite well with that of the experimental QPI at $\mathbf{q}_s$ as shown in Fig. 4g.

## Discussion

We consider the implications of our finding above. Formally, when a material enters into a CDW phase, the Brillouin zone is reduced in size, and the new Bloch states have spectral weight at $\mathbf{k}$ points that are shifted ("folded") by the CDW ordering vector(s) from the normal state $\mathbf{k}$ points. If the CDW is a weak perturbation of the electronic structure (as is shown by the ARPES data in our case), the spectral weight of the folded bands is also small. Quasiparticle scattering can occur between any two Bloch states of the new bandstructure after CDW formation. The formation of the CDW therefore introduces new scattering vectors into a QPI pattern. In particular, scattering vectors will now appear that connect two normal state $\mathbf{k}$ points, but displaced by a CDW vector. It is precisely these scattering vectors that are found above in the STM experiment. However, the intensity of the features in the experiment is unexpectedly strong. Since the intensity of a QPI feature is proportional to the spectral weights of the two states involved in the scattering process, we generally expect that the scattering between normal state $\mathbf{k}$ points will be dominant, while scattering to folded bands is weaker. However, our QPI data shows the complete opposite of this expectation—scattering between normal state hot spots is highly suppressed compared to scattering to the folded hot spots. It is this seeming contradiction between ARPES data (that shows almost no band folding) and STM QPI data (that shows strong features related to band folding) that is our primary result.

In order to resolve this apparent contradiction, we start by analyzing the properties of the electronic Green's function inside the CDW phase. To make the argument more transparent, we consider for simplicity that, in the disordered state, the Green's function $G_0(\omega, \mathbf{k})$ is that of a single-band system with electronic dispersion $\xi_{\mathbf{k}}$, i.e. $G_0^{-1}(\omega,\mathbf{k}) = \omega - \xi_{\mathbf{k}} + i0^+$. Here, $\omega$ denotes frequency and $\mathbf{k}$, momentum. The bond distortions caused by long-range CDW order lead to a new periodicity of the lattice, which in turn acts as a potential energy that folds the band structure by the CDW wave-vector $\mathbf{K}_{CDW}$. Formally, this is described by the self-energy, which, to leading order, is given by $\Sigma(\omega,\mathbf{k}) = \lambda^2 \Delta^2/(\omega - \xi_{\mathbf{k}+\mathbf{K}_{CDW}})$, where $\lambda$ is a coupling constant, and $\Delta$

denotes the order parameter associated with the lattice distortion caused by the CDW. The Green's function inside the CDW phase is then given by $G^{-1}(\omega,\mathbf{k}) = G_0^{-1}(\omega,\mathbf{k}) - \Sigma(\omega,\mathbf{k})$. Consequently, $G$ has poles at the new reconstructed dispersions, which can be approximated by $\xi_{\mathbf{k}}$ and the folded dispersion $\xi_{\mathbf{k}+\mathbf{K}_{CDW}}$ when the coupling $\lambda$ is small. The key point, however, is that the spectral weight of the folded bands is suppressed by a reduced $\lambda$. Since ARPES directly measures the spectral function $-\text{Im}\,G(\omega,\mathbf{k})/\pi$ and sees no clear signature of a folded band structure, this tells us that the coupling constant $\lambda$ is small in $ScV_6Sn_6$. The evolution of the spectral weight of the folded bands on the coupling strength is discussed in detail in the SI (Sections 1.1) and illustrated in Supplementary Fig. 11(a–j).

The tunneling spectrum can also be approximated by $-\text{Im}\,G(\omega,\mathbf{k})/\pi$ in the limit where tunneling matrix elements do not play a strong role. The important difference is that QPI arises due to electrons scattering off impurities present in the sample. As a result, one needs to include the self-energy contribution $\Sigma_{imp} = T$, where the T-matrix $T(\mathbf{k},\mathbf{k}')$ describes the scattering amplitude from the state with momentum $\mathbf{k}$ to the state with momentum $\mathbf{k}'$. The QPI signal $\delta n(\omega, \mathbf{q})$ is dominated by the changes in the DOS due to the impurity scattering; as a result, it is given by

$$\delta n(\omega,\mathbf{q}) = -\frac{1}{\pi}\,\text{Im}\int G(\omega,\mathbf{k})T(\omega;\mathbf{k},\mathbf{k}+\mathbf{q})G(\omega,\mathbf{k}+\mathbf{q})\,d\mathbf{k}. \tag{1}$$

Here, $G(\omega, \mathbf{k})$ and $G(\omega, \mathbf{k} + \mathbf{q})$ correspond to any two distinct Bloch states in the reconstructed CDW phase. In the case of a featureless point-like impurity scattering, which is often assumed in STM experiments, $T(\omega; \mathbf{k}, \mathbf{k}') = T(\omega)$ is momentum-independent (also see discussions on SI, section 1.2). In this situation, $\delta n(\omega, \mathbf{q})$ is dominated by the poles of the Green's function. For the CDW case discussed above, this would give rise to two sets of scattering wavevectors: one set between two normal-state hot-spots and the other set between a normal-state hot-spot and a folded-state hot-spot. This second set of scattering vectors will be suppressed by a factor $\lambda$ relative to the first as discussed above, in direct contradiction to the experiment.

This fundamental discrepancy with ARPES suggests that there is another effect endowing the QPI spectrum with additional momentum dependence. Going beyond a simple one-band description, the real-space structure of the Wannier functions to which the electrons tunnel at the surface is known to impact the QPI dispersion in other compounds[58,59]. To test this idea, we experimentally compared the QPI spectrum obtained from two different termination surfaces, V and $ScSn_2$. As shown in Fig. 4g, the dispersions are identical (for a more detailed comparison, see Supplementary Figs. 6 and 7), suggesting that the real-space structure of the Wannier function is unlikely to govern the observed effect.

Another option is that a different mechanism endows the T-matrix with a non-trivial momentum dependence, $T(\omega; \mathbf{k}, \mathbf{k} + \mathbf{q})$, which in turn is manifested in the QPI. More specifically, from a phenomenological standpoint, if we hypothesize that the T-matrix is strongly peaked at a momentum transfer $\mathbf{q}$ equal to the CDW wave-vector $\mathbf{K}_{CDW}$, the momentum dependence of the T-matrix can compensate for the suppressed spectral weight of the folded electronic states, thus reconciling the QPI and ARPES data (see SI Section 1.3 for a discussion of this point). Of course, this phenomenological description of the data does not address why the T-matrix has such an intrinsic momentum-space profile. We speculate that this effect can arise from interaction-promoted vertex corrections on the underlying impurity potential $V(\mathbf{q})$, from which the T-matrix is derived. Experimentally, strong momentum dependence in scattering has previously been seen in $NbSe_2$[60], and the idea that dynamical fluctuations can dominate STM QPI has previously been invoked in the high-$T_c$ cuprates[61]. Theoretically, previous works have shown that vertex corrections arising from strong magnetic fluctuations endow a featureless impurity potential

with an effective momentum dependence[62,63]. In this regard, we note that in the case of $ScV_6Sn_6$ previous DFT calculations[37,44,45,64] have found that other CDW states are energetically favored, with wavevectors different from the $\tilde{\mathbf{K}} = (1/3, 1/3, 1/3)$ realized experimentally. In particular, the one with the lowest energy has wave-vector $\mathbf{H} = (1/3, 1/3, 1/2)$, which shares the same in-plane components as $\tilde{\mathbf{K}}$. Therefore, one interesting possibility is that the coupling of the electronic states to the fluctuations of this other nearby CDW state could potentially renormalize the impurity potential and make it strongly peaked at the same in-plane $\mathbf{K}_{CDW}$ shared by both the condensed and un-condensed CDW phases. Strong CDW fluctuations have been recently proposed theoretically and observed experimentally in $ScV_6Sn_6$ above the CDW transition[64–66]. Whether they can quantitatively account for the effect proposed here requires further analyses beyond the scope of our work.

We finish by noting that the CDW band folding in $ScV_6Sn_6$ revealed by our QPI measurements has a surprising connection with the electronic structure reconstruction of the $AV_3Sb_5$ compounds, which can be directly seen by ARPES. In the latter, the CDW wave-vector connects the $\bar{\mathbf{M}}$ points where the vHSs are located. This naturally leads to small pockets around these vHS points in the CDW state. It is believed that the interesting emergent phases at low temperature may arise from these small pockets[67,68]. In $ScV_6Sn_6$, one would naively expect that the main impact of the CDW would be in reconstructing the circular pockets around the $\bar{\mathbf{K}}$ points, since they are connected by the CDW wave-vector. Our QPI measurements show that this is not the case. Instead, there is significant band reconstruction near the $\bar{\mathbf{M}}$ point, which is where the QPI hot-spots are located. While there have not so far been reports of additional instabilities in $ScV_6Sn_6$ at low temperatures, these findings, combined with our proposal of residual CDW fluctuations coupled to the electronic degrees of freedom inside the CDW phase, suggest that further measurements looking for emergent electronic phenomena in this compound are warranted.

Note: During the preparation of our manuscript, we noticed two works reporting ARPES and STM studies on $ScV_6Sn_6$[50,69]. The main conclusions of these two papers, including ours, suggest that the physics of the CDW in $ScV_6Sn_6$ is fundamentally different from the canonical kagome metal $AV_3Sb_5$. In all cases, ARPES results suggest a minimal change in the overall electronic structure and the positions of the vHSs across the CDW transition. The STM results of our study and Ref. 50 show striking differences; we see dispersing QPI patterns at different surfaces, whereas such features were not observed in Ref. 50. Furthermore, the tunneling spectra on the V-terminated layer reported in Ref. 50 show an energy gap of 20 meV at the Fermi level, whereas no such gap is observed in our data.

## Methods

### Crystal growth
High-quality single crystals of $ScV_6Sn_6$ were grown using self-flux growth technique from the composition $Sc_{1.1}V_6Sn_{60}$. The starting elements Sc (Ames Lab), V (99.9% pure from Sigma Aldrich), and Sn (99.9999 + % pure from Alfa Aesar) were loaded into an alumina crucible inside an Ar-filled glove box and were vacuum-sealed in a quartz tube. The assembly was heated to 1150 °C at a rate of 50 °C/h and held at that temperature for 15 h. It was then cooled to 780 °C at a rate of 1 °C/h and the single crystals were obtained by extracting the molten flux using a centrifuge. The sample homogeneity and chemical composition were confirmed using a JEOL scanning electron microscope equipped with an EDS (energy-dispersive x-ray spectroscopy) analyzer. The EDS measurements yielded a composition $ScV_{5.98(5)}Sn_{6.00(5)}$, which is the stoichiometric composition to within the error bars.

### Scanning near-field optical microscopy
The nano-infrared scattering experiments were performed using a home-built cryogenic scattering-type scanning near-field optical microscope (s-SNOM) housed in an ultra-high vacuum chamber with a base pressure of ~$7 \times 10^{-11}$ torr. The cryogenic s-SNOM enables imaging of surface optical properties at variable temperatures with nanoscale resolution. The s-SNOM works by scattering tightly focused light from a sharp atomic force microscope (AFM) tip. First, the middle-infrared light from a quantum cascade laser was focused onto a metalized AFM tip using a parabolic mirror. Then the back-scattered light was registered and demodulated using the pseudoheterodyne method. With this approach, we obtained the genuine near-field signal with a resolution of ~20 nm. In this work, we present images of the back-scattered near-field amplitude demodulated at the fourth harmonic (abbreviated to nano-IR scatting amplitude, or $s_4$. We recorded nano-IR scattering amplitude images on a natural crystal facet from room temperature across the CDW phase transition down to 86 K. There is a temperature sensor underneath the sample holder to read out the variable temperature. In addition, the temperature of the measured crystal is further calibrated by the reported CDW phase transition temperature.

### STM/STS measurements
STM data were acquired at ~6.7 K. Single crystals of $ScV_6Sn_6$ are pre-cooled and quickly cleaved in ultra-high vacuum before being inserted into the STM scanner. A chemically etched tungsten tip was annealed and calibrated on the Au(111) surface. Spectroscopic data were taken by the standard lock-in technique at 2.451 kHz with bias voltage applied to the sample. Real-space DOS images were acquired by the demodulation of the tunneling current when a small excitation bias voltage was applied with the tip scanning over the sample surface in a constant DC current mode. Symmetrized FT analyses were performed on the real-space DOS images to obtain the FT-STS images at each energy. To enhance the signal-to-noise ratio of FT-STS images, we performed symmetrization ($C_6$ and mirror symmetry) using the Bragg peaks.

### ARPES measurements
ARPES experiments were performed at the Electron Spectro-Microscopy (ESM) 21-ID-1 beamline of the National Synchrotron Light Source II, USA. The beamline is equipped with a Scienta DA30 electron analyzer, with base pressure better than ~$1 \times 10^{-11}$ mbar. Prior to the ARPES experiments, samples were cleaved inside an ultra-high vacuum chamber (UHV) at ~10 K and 120 K for the V- and $ScSn_2$-terminated samples, respectively. The total energy and angular resolution were ~12 meV and ~0.1°, respectively. All measurements were performed using linearly horizontal (LH) polarized light.

### Density functional theory (DFT) calculations
DFT calculations were performed using the linear augmented plane wave (LAPW) method and local-density approximations (LDA) functional, as implemented in Wien2k version 14.2[70]. For self-consistent electronic structure calculations, we used a $20 \times 20 \times 10$ $\Gamma$-centered k-point mesh with RKmax set to 8.5, and experimental lattice parameters as reported in Ref. 41.

### Reporting summary
Further information on research design is available in the Nature Portfolio Reporting Summary linked to this article.

## Data availability
The data that supports the plots within this paper and other findings of this study have been deposited in the Zenodo database (https://doi.org/10.5281/zenodo.11153103). All other data used in this study are available from the corresponding author upon request.

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

## Acknowledgements

The STM measurements and nano-optics measurements were supported by Programmable Quantum Materials, an Energy Frontier Research Center funded by the U.S. Department of Energy (DOE), Office of Science, Basic Energy Sciences (BES), under award DE-SC0019443. ARPES measurements used resources at 21-ID (ESM) beamline of the National Synchrotron Light Source II, a U.S. Department of Energy (DOE) Office of Science User Facility operated for the DOE Office of Science by Brookhaven National Laboratory under Contract No. DE-SC0012704. Salary support was also provided by the National Science Foundation via grants DMR-2004691 (A.N.P.) and DMR-2011738 (S.S.). The research at Ames National Laboratory was supported by the U.S. Department of Energy, Office of Basic Energy Sciences, Division of Materials Sciences and Engineering. Ames National Laboratory is operated for the U.S. Department of Energy by Iowa State University under Contract No. DE-AC02-07CH11358. T.V. was supported by the Gipuzkoa Provincial Council, grant 2023-CIEN-000046-01. E.R. and T.B. were supported by the NSF CAREER grant DMR-2046020. R.M.F. was supported by the Air Force Office of Scientific Research under Award No. FA9550-21-1-0423.

## Author contributions

A.K.K. and T.V. performed the ARPES measurements and analyzed the data. T.Y. and E.V. provided technical assistance for the ARPES measurements. The STM measurements and data analysis were performed by X.H. and E.S. with the help of S.T. and S.S. Also, E.R., T.B., and R.M.F. provided theoretical input for DFT calculation and modeling. S.Z. performed the nano-IR measurements with help from D.S. under the supervision of D.N.B. and C.R.D. S.P. grew the bulk single-crystals and characterized the sample stoichiometry using EDS under the supervision of D.C.J. A.K.K., X.H., R.M.F., and A.N.P. wrote the paper with input from all authors. All authors reviewed the paper. A.N.P. supervised the project.

## Competing interests

The authors declare no competing interests.
