## [Peer Review File · Nature Communications]

Low-Energy Electronic Structure in the Unconventional Charge-Ordered State of ScV₆Sn₆REVIEWER COMMENTS

Reviewer #1 (Remarks to the Author):

Kundu et al present impressive PES and STM measurement of ScV6Sb6 in deep in the CDW phase. Data quality and coverage of key phenomena is high. Kundu shows by clear argumentation that PES and STM data of CDW at face value are not internally consistent. To this point a nice result.

Kundu then describes the Green's functions of electrons in the CDW phase as if the CDW is a disordered state with self $\Sigma(\omega, k) = \lambda^2 \Delta^2 / (\omega - \xi_k + K_{CDW})$, where λ is the coupling constant and Δ the order parameter of CDW. Kundu then proposes that the Greens function retains the poles that has poles of the metallic state ξ_k before the CDW formed.

This is an extraordinary proposal. The textbook description is radically different: when the crystal lattice symmetry changes due to the CDW, the Hamilton is diagonalized for states in the new BZ at the correct chemical potential satisfying Luttinger for the new unit cell, and the all the poles represent new eigenstates for whom, at $T=0$, $\Sigma(\omega, k)=0$.

Kundu nicely describes the impurity scattering QPI process based on a scattering matrix $T(\omega; k, k + q)$ but then, in order it seems to explain the q-space symmetry of the QPI data, represents the CDW itself as an impurity potential whose T matrix is strongly peaked at momenta of CDW wavevectors. Again this is unrecognizable in elementary CDW theory and indeed violated Bloch's Theorem for eigenstates of the new Unit-cell / Brillouin-zone.

The Kundu hypothesis that the CDW is nothing more than an impurity in the original metallic state is not physically impossible in the case where the CDW is undergoing a transition, or has no long range order, or its order parameter is dominated by disorder. But no direct experimental evidence for such situations is put forward.

Also there is a much more straightforward potential explanation for the symmetry of the QPI data, it is that the localized Wannier functions of the crystal terminator layer through which tunneling occurs have a different symmetry than that of the bulk band structure e.g. Phys. Rev. B 96, 174523 (2017).

Reviewer #2 (Remarks to the Author):

Kundu et al. present a combined ARPES and STM study, with additional near-field optic data and DFT simulations, of ScV6Sn6. A highly topical Kagome system.

In terms of novelty, and interest to the community, I find the data presents a detailed picture of the low energy electronic structure of this new and exciting system. The data meets the criteria for nature communications.

However, I cannot recommend publication of this manuscript in its current form.

The wider story attached to this data, that "strong fluctuations" are present and responsible for the observed QPI pattern, is based on an incorrect assumption. Once this logical fallacy is realized the arguments about "strong fluctuations in the charge ordered state", "significant residual electronic interactions", "interaction-dressed impurity scatters" or T-matrices with a strong momentum dependence becomes speculation without evidence.

The error lies in assuming that the joint density of states (JDOS) – or convolution of the spectral function - should reproduce the experimental QPI pattern.

"We see that the corresponding JDOS completely fails to reproduce the prominent dispersing spots at as seen experimentally.... We must therefore look for explanations beyond the simple model for QPI presented above"

There are many examples in the literature of JDOS simulations not reproducing QPI intensities (see for example Fig. 11 in Ref. [R1] or Fig. 6 in Ref. [R2]) and from a mathematical point of view they won't reproduce this, even in the Born limit discussed in this manuscript. This is because the JDOS is the autocorrelation of the imaginary part of the Green's function, whereas QPI is the imaginary part of the convolution of the Green's function.

The intensity of QPI scattering vectors is a consequence of the real-space intra-unit cell structure, which is excluded in momentum space only models. One needs to take a continuum description of the electronic structure, involving appropriate Wannier orbitals of the local atomic basis, and perform a full T-matrix calculation to correctly reproduce these intensities. This has been done for e.g the Cuprates (Ref. [R3]), Sr₂RuO₄ (Ref. [R4]) and other systems.

It is also important that one chooses the correct unit cell – The authors here assume a unit cell that does not have the CDW reconstruction, as evidenced by the lack of backfolding in the ARPES data, however the STM topographies show a very strong CDW unit cell which would modify the QPI. If this CDW basis is chosen for their model and a continuum T-matrix calculation is performed, then one would be able to simulate the CDW bragg peaks and any scattering vectors associated with them, as well as their intensities. This change of basis would also fold the scattering vectors from the hotspot model and offer a much simpler explanation for their observation, without needing to invoke momentum dependent T-matrices.

In short there are more trivial explanations for the discrepancy between the simulation and experiment that the authors have not ruled out.

My final criticism, is that the conclusions drawn about the need for "strong correlations" are not supported by any additional simulation. The authors propose that a momentum dependent T-matrix, caused by "strong electronic fluctuations" (eq. 1) is necessary to explain the discrepancy between simulation and experiment, but do not calculate such a quantity to show that this would address the issue. Therefore rendering it a hypothesis rather than a result. This simulation should not be that difficult, T-Matrix simulations from DFT derived models are routinely performed in the literature [see e.g R1-R4]. Without that simulation to support their claim, then the titular conclusion, "Quasiparticle interference reveals strong fluctuations in the Charge Order State of ScV₆Sn₆" has not been demonstrated.

I wish to stress that the ARPES, STM and optical data is of very high quality and provides important information into this new Kagome material. Highlighting the different consequences the CDW has on the electronic structure compared to the well known AV₃Sb₅ systems. Nevertheless the manuscript presents a narrative that has not been proved, and reading this one would be mistaken for assuming that exotic phenomena is present even when many trivial mechanisms could equally explain the data. I therefore do not recommend publication unless sufficient revisions are made to the discussion and conclusions of this manuscript.

As an aside there have also been similar ARPES and STM studies appearing recently on the ArXiv (Ref [R5, R6], (although no QPI), which I'm sure the authors are aware of. It would be of interest to readers if the authors could mention whether this manuscript supports or contradicts these studies e.g. as a note in a revised edition of the manuscript.

References

- [R1] Derry et al. PRB 92, 035126 (2015)
- [R2] Kohsaka et al. PRB 95, 115307 (2017)
- [R3] Kreisel et al. PRL, 114, 217002 (2015)
- [R4] Kreisel et al. npj Quantum Materials, 6, 100 (2021)
- [R5] Lee et al. arXiv::2304.11820 (2023)
- [R6] Cheng et al. arXiv:2302.12227 (2023)

Response to the comments about the submitted paper: *Quasiparticle Interference Reveals Strong Fluctuations in the Charge-Ordered State of ScV_6Sn_6*

We thank the referees for the time and effort they spent reviewing our paper, which have helped us improve our presentation of data in the revised version. In the following point-by-point response, the reviewers' comments are italicized, while our responses are in regular font.

Answers to Reviewer 1

Comment R1.1 *Kundu et al present impressive PES and STM measurement of ScV6Sb6 in deep in the CDW phase. Data quality and coverage of key phenomena is high. Kundu shows by clear argumentation that PES and STM data of CDW at face value are not internally consistent. To this point a nice result.*

Answer to R1.1 We thank the reviewer for appreciating the quality of our data which is ultimately the basis of our experimental work.

Comment R1.2 *Kundu then describes the Green's functions of electrons in the CDW phase as if the CDW is a disordered state with self $\Sigma(\omega, \mathbf{k}) = \lambda^2 \Delta^2 / (\omega - \xi_{\mathbf{k}+\mathbf{K}_{\text{CDW}}})$, where λ is the coupling constant and Δ the order parameter of CDW. Kundu then proposes that the Greens function retains the poles that has poles of the metallic state $\xi_{\mathbf{k}}$ before the CDW formed. This is an extraordinary proposal. The textbook description is radically different: when the crystal lattice symmetry changes due to the CDW, the Hamilton is diagonalized for states in the new BZ at the correct chemical potential satisfying Luttinger for the new unit cell, and the all the poles represent new eigenstates for whom, at $T=0$, $\Sigma(\omega, \mathbf{k})=0$*

Answer to R1.2 We believe there is a misunderstanding here. The expression that we wrote is in fact the standard textbook expression for the electronic propagator in the presence of long-range CDW order. After inserting the self-energy $\Sigma = \frac{\lambda^2 \Delta^2}{\omega - \xi_{\mathbf{k}+\mathbf{K}_{\text{CDW}}}}$ in the Dyson's equation $G^{-1} = G_0^{-1} - \Sigma$, with $G_0^{-1} = \omega - \xi_{\mathbf{k}}$, the retarded Green's function inside the CDW phase is given by:

$$\begin{aligned} G^{-1}(\omega, \mathbf{k}) &= \omega - \xi_{\mathbf{k}} - \frac{\lambda^2 \Delta^2}{\omega - \xi_{\mathbf{k}+\mathbf{K}_{\text{CDW}}}} \\ &= \frac{(\omega - E_{+, \mathbf{k}})(\omega - E_{-, \mathbf{k}})}{\omega - \xi_{\mathbf{k}+\mathbf{K}_{\text{CDW}}}} \end{aligned} \quad (1)$$

where, as usual for retarded functions, ω should be understood as $\omega + i0^+$, and we defined:

$$E_{\pm, \mathbf{k}} \equiv \left(\frac{\xi_{\mathbf{k}} + \xi_{\mathbf{k}+\mathbf{K}_{\text{CDW}}}}{2} \right) \pm \sqrt{\left(\frac{\xi_{\mathbf{k}} - \xi_{\mathbf{k}+\mathbf{K}_{\text{CDW}}}}{2} \right)^2 + \lambda^2 \Delta^2} \quad (2)$$

Thus, the Green's function has poles at $E_{\pm, \mathbf{k}}$, which correspond to the standard expressions for the electronic dispersion inside the CDW phase. Indeed, the dispersion can be obtained by superimposing the unfolded band $\xi_{\mathbf{k}}$ and the folded band $\xi_{\mathbf{k}+\mathbf{K}_{\text{CDW}}}$, with a gap proportional to $\lambda\Delta$ opening at the intersections between them. As expected, $E_{\pm, \mathbf{k}} = E_{\pm, \mathbf{k}+\mathbf{K}_{\text{CDW}}}$, which reflects the fact that the Brillouin zone in the CDW phase is the Brillouin zone in the non-ordered phase folded by the ordering vector \mathbf{K}_{CDW} .

To make these points clear to the reader, we have added the discussion above to Supplementary Information section 1.1 and refer to it in the main text on Page 10.

Comment R1.3 *Kundu nicely describes the impurity scattering QPI process based on a scattering matrix $T(\omega; k, k+q)$ but then, in order it seems to explain the q -space symmetry of the QPI data, represents the CDW itself as an impurity potential whose T matrix is strongly peaked at momenta of CDW wavevectors. Again this is unrecognizable in elementary CDW theory and indeed violated Bloch's Theorem for eigenstates of the new Unit-cell / Brillouin-zone.*

Answer to R1.3 As with the previous question, we believe this is another misunderstanding. The electronic dispersion inside the CDW phase is described by the usual Green's function, Eq. (1) above. The point we are making is that if the coupling constant λ is very small, the spectral weight of the folded bands in Eq. (2) will generally be negligible. This can be seen explicitly by rewriting Eq. (1) as:

$$G(\omega, \mathbf{k}) = \frac{W_{+, \mathbf{k}}}{\omega - E_{+, \mathbf{k}}} + \frac{W_{-, \mathbf{k}}}{\omega - E_{-, \mathbf{k}}} \quad (3)$$

where W_{\pm} , the spectral weights of the poles, are given by:

$$W_{\pm, \mathbf{k}} = \frac{1}{2} \left(1 \pm \frac{\text{sgn}(\xi_{\mathbf{k}} - \xi_{\mathbf{k} + \mathbf{K}_{\text{CDW}}})}{\sqrt{1 + \left(\frac{2\lambda\Delta}{\xi_{\mathbf{k}} - \xi_{\mathbf{k} + \mathbf{K}_{\text{CDW}}}} \right)^2}} \right) \quad (4)$$

To proceed, we define $\delta_{\mathbf{k}} \equiv (\xi_{\mathbf{k}} - \xi_{\mathbf{k} + \mathbf{K}_{\text{CDW}}})/2$ and consider any momenta \mathbf{k} for which $|\delta_{\mathbf{k}}| \gg \lambda\Delta$. For small coupling λ , this condition is met for a wide range of momenta for which the folded and unfolded bands do not cross. Expanding to leading order in $\lambda\Delta$ then gives:

$$\begin{aligned} E_{\text{sgn}(\delta), \mathbf{k}} &\approx \xi_{\mathbf{k}} \\ E_{-\text{sgn}(\delta), \mathbf{k}} &\approx \xi_{\mathbf{k} + \mathbf{K}_{\text{CDW}}} \\ W_{\text{sgn}(\delta), \mathbf{k}} &\approx 1 - \frac{\lambda^2 \Delta^2}{4\delta_{\mathbf{k}}^2} \\ W_{-\text{sgn}(\delta), \mathbf{k}} &\approx \frac{\lambda^2 \Delta^2}{4\delta_{\mathbf{k}}^2} \end{aligned} \quad (5)$$

such that the Green's function inside the CDW phase becomes:

$$G(\omega, \mathbf{k}) \approx \left(1 - \frac{\lambda^2 \Delta^2}{4\delta_{\mathbf{k}}^2} \right) \frac{1}{\omega - \xi_{\mathbf{k}}} + \left(\frac{\lambda^2 \Delta^2}{4\delta_{\mathbf{k}}^2} \right) \frac{1}{\omega - \xi_{\mathbf{k} + \mathbf{K}_{\text{CDW}}}} \quad (6)$$

Clearly, for these momenta values, the spectral weight of the folded bands is negligible. This is what we believe happens in our system, since ARPES, which directly measures the electronic spectral weight $-\frac{1}{\pi} \text{Im} G(\omega, \mathbf{q})$, does not “see” the folded bands.

Now, the QPI signal $\delta n(\omega, \mathbf{q})$ depends not only on the electronic spectral weight, but also on the impurity potential encoded in the T -matrix:

$$\delta n(\omega, \mathbf{q}) = -\frac{1}{\pi} \text{Im} \int G(\omega, \mathbf{k}) T(\omega; \mathbf{k}, \mathbf{k} + \mathbf{q}) G(\omega, \mathbf{k} + \mathbf{q}) d\mathbf{k} \quad (7)$$

Using Eq. (6), it is clear that for a featureless T -matrix, $T(\omega; \mathbf{k}, \mathbf{k} + \mathbf{q}) = T_0(\omega)$, corresponding e.g. to a point-like impurity potential, the QPI inside the CDW ordered state is well approximated by that of the disordered phase, $\delta n(\omega, \mathbf{q}) \approx \delta n_0(\omega, \mathbf{q}) + \mathcal{O}(\lambda^2 \Delta^2)$, where we defined

$$\delta n_0(\omega, \mathbf{q}) = -\frac{1}{\pi} T_0(\omega) \text{Im} \int G_0(\omega, \mathbf{k}) G_0(\omega, \mathbf{k} + \mathbf{q}) d\mathbf{k} \quad (8)$$

However, this contradicts our experimental QPI, which shows clear signatures of the folded band. This could be explained if the T -matrix also had a component that is strongly peaked at the

CDW wave-vector \mathbf{K}_{CDW} . To illustrate this point, consider the following function: $T(\omega; \mathbf{k}, \mathbf{k} + \mathbf{q}) = T_0(\omega) [1 + \alpha \delta(\mathbf{q} - \mathbf{K}_{\text{CDW}})]$. The resulting QPI is now:

$$\delta n(\omega, \mathbf{q}) \approx \delta n_0(\omega, \mathbf{q}) - T_0(\omega) \frac{\alpha}{\pi} \delta(\mathbf{q} - \mathbf{K}_{\text{CDW}}) \text{Im} \int G_0(\omega, \mathbf{k}) G_0(\omega, \mathbf{k} + \mathbf{K}_{\text{CDW}}) d\mathbf{k} + \mathcal{O}(\lambda^2 \Delta^2) \quad (9)$$

The middle term contains both the folded and unfolded bands, but is independent of the coupling constant λ . Importantly, this does not imply that the CDW is described by an impurity potential. Quite on the contrary, as explained above, the Green's function in the CDW state is the standard one. What this assumption means is that there is some interaction that dresses the featureless point-like impurity potential, rendering it strongly momentum-dependent. Our proposal is that fluctuations of the un-condensed competing CDW state, which is peaked at the same in-plane wave-vector \mathbf{K}_{CDW} , are responsible for these interactions. Indeed, it has been shown previously that bosonic fluctuations dress the impurity potential in such a way as that it acquires a non-trivial momentum dependence [see Refs. 62, 63]. Of course, there could be other possible sources for this effect, such as non-point-like impurities. But our key point is that there is a seemingly disagreement between the ARPES and STM data on the same samples.

We have added the discussion above to Supplementary Information sections 1.1 and 1.2 and have referred to it in the main text on Page 10 of the manuscript, where we have added the text to read “The evolution of the spectral weight of the folded bands on the coupling strength is discussed in detail in the SI (sections 1.1 and 1.2) and illustrated in Supplementary Fig. 11(a)-(j).”.

To further help the reader visualize the effect of the CDW on the band folding and on the QPI, we have performed a simple calculation (added to new Supplementary Information section 1.1 and 1.2) to illustrate how the intensity of the folded bands and the QPI features depend on the coupling strength (see Fig. R1 below and Supplementary Fig. 11). We work with a nearest neighbor tight-binding model on a triangular lattice. The energy-momentum dispersion relation can be written as

$$E(k_x, k_y) = -2t + 4t \cos(k_x a/2) [\cos(k_x a/2) + \cos(\sqrt{3} k_y a/2)] \quad (10)$$

We then impose a CDW with a wavevector of $(1/3, 1/3)$ of the Bragg peak, similar to the case of ScSn_6V_6 . We vary the coupling strength of the CDW to the electronic band structure, $\frac{\lambda \Delta}{t} = 0, 0.05, 0.1, 0.2, 0.5$. Shown in Fig. R1a-e, is the spectral function (which ARPES directly probes) along the k_x direction plotted in units of the hopping energy t for the various choices of the coupling strength. It is seen clearly that as the coupling strength increases, the intensity of the folded bands and the energy gaps at the crossing points of the folded and unfolded bands also increases. In Fig. R1f-j, the Fermi surface is shown with a 6-fold fermi pocket centered at Γ in the Brillouin zone (for a specific choice of the chemical potential, $\mu = 1.4t$). Once again, the folded bands become more evident as the coupling strength of the CDW is increased.

Finally, we calculate the JDOS, the autocorrelation of the spectral function, at the Fermi level. The result is plotted in Fig. R1k-o. While there are several features in the JDOS, the scattering vectors of relevance to our experiment are those that are close to the CDW wave vector, which correspond to scattering between the bare and folded bands. A zoom in of the JDOS around one of the CDW wave vectors is shown in Fig. R1p-t. It is evident that the intensity of the scattering between the bare and folded bands increases proportionally with the spectral weight of the folded

Fig. R1: Lattice distortion and its coupling to the electronic band structure. **a-e**, spectral function along $\Gamma - K$ direction for different coupling strengths of the CDW ($\frac{\lambda\Delta}{t} = 0, 0.05, 0.1, 0.2, 0.5$). The CDW wavevector is chosen to be $\mathbf{K}_{\text{CDW}} = (1/3, 1/3)$ of the original Bragg point. **f-j**, corresponding Fermi surface intensity of the spectral function (with $\mu = 1.4t$ marked by the green dashed line in **(a)**), **k-o**, JDOS at the Fermi level, and, **p-t** zoom-in of the JDOS around \mathbf{K}_{CDW} , respectively. The band reconstruction effects (spectral weight for the folded band and gap opening) become enhanced as the coupling strength is increased. The intensity of the JDOS around \mathbf{K}_{CDW} (region enclosed by the circles) that corresponds to scattering between the bare and folded bands becomes pronounced as the CDW strength is increased. Blue hexagons mark the first Brillouin zone boundary.

bands.

Comment R1.4 *The Kundu hypothesis that the CDW is nothing more than an impurity in the original metallic state is not physically impossible in the case where the CDW is undergoing a transition, or has no long range order, or its order parameter is dominated by disorder. But no direct experimental evidence for such situations is put forward.*

Answer to R1.4 We agree with the reviewer that there is no direct experimental evidence other than the QPI in our manuscript. We would like to point out that the “strong fluctuations” and “T-matrices with a strong momentum dependence” in the CDW state that we wrote in the manuscript is our hypothesis. We found this hypothesis can better explain our QPI data than other possibilities such as the localization of the Wannier function as discussed below. In the revised version we clearly mentioned that this is our hypothesis and revised the text accordingly. We have modified the title of our paper to avoid misleading readers in this regard. In the abstract, we modified the text to read ‘...Thus, the observed QPI is inconsistent, at face value, with the observed ARPES spectra. As a plausible explanation for this discrepancy, we propose the presence of a strong momentum-dependent scattering potential peaked at the CDW wave-vector deep within the ordered phase...’. On Page 11, we modified the text to read ‘...This fundamental discrepancy with ARPES suggests that there is another effect endowing the QPI spectrum with additional momentum dependence...’, ‘...Another option is that a different mechanism endows the T -matrix with a non-trivial momentum dependence, $T(\omega; \mathbf{k}, \mathbf{k} + \mathbf{q})$, which in turn is manifested in the QPI. More specifically, from a phenomenological standpoint, if we hypothesize that the T -matrix is strongly peaked at a momentum transfer q equal to the CDW wave-vector \mathbf{K}_{CDW} , the momentum dependence of the T -matrix can compensate for the suppressed spectral weight of the folded electronic states, thus reconciling the QPI and ARPES data...’. On Page 12, we modified the text to read ‘...Of course, this phenomenological description of the data does not address why the T -matrix has such an intrinsic momentum-space profile...Instead, here we speculate that...’.

We want to note that the recent theoretical and experimental studies show evidence of strong CDW fluctuations above T_{CDW} in ScV_6Sn_6 due to the presence of nearly degenerate CDW configurations [see Refs. 64–66]. A recent theoretical study [64] also highlights the possibility of transition from the $(\sqrt{3} \times \sqrt{3} \times 3)$ to the $(\sqrt{3} \times \sqrt{3} \times 2)$ phase at low temperatures due to reduced energy barrier and warrants further experimental studies. If such a transition happens in reality, then one would also expect strong fluctuations near this phase transition at low temperatures.

Comment R1.5 *Also there is a much more straightforward potential explanation for the symmetry of the QPI data, it is that the localized Wannier functions of the crystal terminator layer through which tunneling occurs have a different symmetry than that of the bulk band structure e.g. Phys. Rev. B 96, 174523 (2017)*

Answer to R1.5 We agree with the reviewer that the real-space structure of the localized Wannier functions at the surface could be different than bulk. One way to assess this effect experimentally is via different surface terminations. We expect on general grounds that the surface orbitals are different on different surface terminations, and therefore the filtering effect of the Wannier functions is also different. Therefore, taking data on different surface terminations can help us address this question experimentally. In our experiment, we are able to find surfaces terminated by Vanadium, as well as ScSn_2 . We have now taken large data sets on both surfaces and are able to directly

compare the QPI from each surface. While there are changes in the intensity of features on the two surfaces, all of the features that we discuss in our experiment are completely reproduced on both surfaces (see Fig. R2 below). This suggests to us that the real-space structure of the Wannier function does not play the dominant role in determining the QPI of our system. This is also in line with our ARPES Fermi surface maps where most of the Fermi surface features are captured reasonably well by the bulk calculations.

Fig. R2: QPI maps on V- and ScSn₂ surfaces. **a** and **c** FT-STIS image near the Fermi level for V- and ScSn₂ surfaces, respectively. Green and red dots mark the positions of Bragg and CDW peaks, respectively. The blue circle highlights the prominent QPI spots surrounding the CDW peaks. **b**, Schematic drawing of the QPI features shown in (a and c). The most prominent QPI features (at the momentum q_s indicated by the pink arrow) are 3-fold symmetric with respect to the CDW peaks. **c** QPI dispersion from two different surfaces (V and ScSn₂) together with q_1 from simulation.

To make these points clear to the reader, we have added discussions of the QPI on different surface terminations in the main text and also in Supplementary Information. On Page 11 of the manuscript, we have modified the text to read ‘...This fundamental discrepancy with ARPES suggests that there is another effect endowing the QPI spectrum with additional momentum dependence. Going beyond a simple one-band description, the real-space structure of the Wannier

functions to which the electrons tunnel at the surface is known to impact the QPI dispersion in other compounds [58,59]. To test this idea, we experimentally compared the QPI spectrum obtained from two different termination surfaces, V and ScSn₂. As shown in Fig. 4g, the dispersions are identical (for a more detailed comparison, see Supplementary Figs. 6 and 7), suggesting that the real-space structure of the Wannier function is unlikely to govern the observed effect...’.

Finally, we note that the reliable Wannierization for this multi-band system is challenging. The challenges are discussed in details in our response to Reviewer 2 (Answer to R2.4). However, this definitely warrants future theoretical study.

Answers to Reviewer 2

Comment R2.1 *Kundu et al. present a combined ARPES and STM study, with additional near-field optic data and DFT simulations, of ScV_6Sn_6 . A highly topical Kagome system. In terms of novelty, and interest to the community, I find the data presents a detailed picture of the low energy electronic structure of this new and exciting system. The data meets the criteria for nature communications. However, I cannot recommend publication of this manuscript in its current form.*

Answer to R2.1 We thank the reviewer for finding our results novel and interesting and at the same time raising some concerns regarding the interpretation of the data. In the following, we have responded to all the questions raised by the reviewer and modified the manuscript accordingly.

Comment R2.2 *The wider story attached to this data, that “strong fluctuations” are present and responsible for the observed QPI pattern, is based on an incorrect assumption. Once this logical fallacy is realized the arguments about “strong fluctuations in the charge ordered state”, “significant residual electronic interactions”, “interaction-dressed impurity scatters” or T -matrices with a strong momentum dependence becomes speculation without evidence.*

Answer to R2.2 Similar to our response to the reviewer 1, we would like to point out that the *strong fluctuations* and *T -matrices with a strong momentum dependence* in the CDW state that we wrote in the manuscript is our hypothesis. We found this hypothesis can better explain our QPI data than other possibilities such as the real-space structure of the Wannier function as discussed above. In the revised version we clearly mentioned that this is our hypothesis. We have modified the title of our paper to avoid misleading readers in this regard. In the abstract, we modified the text to read ‘...Thus, the observed QPI is inconsistent, at face value, with the observed ARPES spectra. As a plausible explanation for this discrepancy, we propose the presence of a strong momentum-dependent scattering potential peaked at the CDW wave-vector deep within the ordered phase...’. On Page 11, we modified the text to read ‘...This fundamental discrepancy with ARPES suggests that there is another effect endowing the QPI spectrum with additional momentum dependence...’, ‘...Another option is that a different mechanism endows the T -matrix with a non-trivial momentum dependence, $T(\omega; \mathbf{k}, \mathbf{k} + \mathbf{q})$, which in turn is manifested in the QPI. More specifically, from a phenomenological standpoint, if we hypothesize that the T -matrix is strongly peaked at a momentum transfer \mathbf{q} equal to the CDW wave-vector \mathbf{K}_{CDW} , the momentum dependence of the T -matrix can compensate for the suppressed spectral weight of the folded electronic states, thus reconciling the QPI and ARPES data...’. On Page 12, we modified the text to read ‘...Of course, this phenomenological description of the data does not address why the T -matrix has such an intrinsic momentum-space profile...Instead, here we speculate that...’.

Comment R2.3 *The error lies in assuming that the joint density of states (JDOS) – or convolution of the spectral function - should reproduce the experimental QPI pattern. We see that the corresponding JDOS completely fails to reproduce the prominent dispersing spots as seen experimentally. . . . We must therefore look for explanations beyond the simple model for QPI presented above”. There are many examples in the literature of JDOS simulations not reproducing QPI intensities (see for example Fig. 11 in Ref. Derry et al. PRB 92, 035126 (2015) or Fig. 6 in Ref. Kohsaka et al. PRB 95, 115307 (2017)) and from a mathematical point of view they won’t reproduce this, even in the Born limit discussed in this manuscript. This is because the JDOS is*

the autocorrelation of the imaginary part of the Green's function, whereas QPI is the imaginary part of the convolution of the Green's function.

Answer to R2.3 We agree with the reviewer that from a mathematical point of view, the JDOS and QPI are different. The JDOS is proportional to $\int \text{Im}(G_0(\omega, \mathbf{k}))\text{Im}(G_0(\omega, \mathbf{k} + \mathbf{q}))d\mathbf{k}$ while the QPI is proportional to $\text{Im}\int G_0(\omega, \mathbf{k})G_0(\omega, \mathbf{k} + \mathbf{q})d\mathbf{k}$.

There are some systems as pointed out by the reviewer, where the JDOS simulation does not reproduce the QPI features. However, the point we are making is simple: when the CDW order parameter is weak, the effect of CDW folding on both the real and imaginary part of the Green's function is also correspondingly weak (see Eq. 6 in response to comment 1.3). Therefore, we expect both the JDOS and QPI to be small near \mathbf{K}_{CDW} based on the absence of clear folding in ARPES. This point can also be demonstrated with the simple 2D tight binding model described in the response to comment 3 from referee 1.

To make this point clearer to the reader, on Page 7 of the manuscript, we have modified the text to read “While the formal definition of the QPI is different from the JDOS (See Section 1.2 in the Supplementary Information for discussion on this), we expect that when the spectral intensity of the folded bands is weak, both the JDOS and QPI are small for scattering that involves the folded bands.”

Comment R2.4 *The intensity of QPI scattering vectors is a consequence of the real-space intra-unit cell structure, which is excluded in momentum space only models. One needs to take a continuum description of the electronic structure, involving appropriate Wannier orbitals of the local atomic basis, and perform a full T-matrix calculation to correctly reproduce these intensities. This has been done for e.g the Cuprates (Ref. Kreisel et al. PRL, 114, 217002 (2015), Sr2RuO4 (Kreisel et al. npj Quantum Materials, 6, 100 (2021)) and other systems.*

Answer to R2.4 As we have discussed in the response to referee 1, a practical way to consider the effect of the real-space structure of the Wannier functions at the surface is to consider different surface terminations involving different atoms. We now have extended data sets on both the vanadium and ScSn₂ surfaces showing identical QPI features, indicating to us that the real-space structure of the Wannier function does not play the dominant role in the observed QPI of our system.

Theoretically, we note that there are several subtleties about the Wannierization procedure and the nature of the Wannier orbitals that make such an attempt very challenging in the vanadium based kagome compounds. First, and perhaps most importantly, these are multiband systems unlike, for example, the cuprates. As a result, the Wannier basis is not uniquely defined, and there is only so much information one can extract from the localization, shape, and symmetry of the Wannier orbitals. (The ‘maximally localized’ Wannier orbitals are unique, however, there is no reason to ascribe them more priority than any other set of Wannier functions.) Additionally, on the practical side, it is extremely difficult to obtain a Wannier model for the Vanadium based kagome metals in a slab configuration. For example, in the Kreisel et al. study that the referee mentions, a single layer of cuprates was considered sufficient to build a Wannier model. However, in our system, there are many bands that are dispersive along the c-axis as well, and a realistic Wannier model would require many atomic layers which comes with additional computational cost and complexity. Additionally, the less ionic nature of our compound (which can be roughly predicted by the small electronegativity difference between V and Sb, compared to that between Cu and O) will make the shape and size of the Wannier orbitals more complicated, and hence

will make both the qualitative analysis of the Wannier-based tight-binding Hamiltonian and a T-matrix calculation practically impossible.

As discussed in the response to the referee 1, we have added discussions with QPI on different surface terminations in the main text and also in Supplementary Information. On Page 11 of the manuscript, we have modified the text to read “...This fundamental discrepancy with ARPES suggests that there is another effect endowing the QPI spectrum with additional momentum dependence. Going beyond a simple one-band description, the real-space structure of the Wannier functions to which the electrons tunnel at the surface is known to impact the QPI dispersion in other compounds [58, 59]. To test this idea, we experimentally compared the QPI spectrum obtained from two different termination surfaces, V and ScSn₂. As shown in Fig. 4g, the dispersions are identical (for a more detailed comparison, see Supplementary Figs. 6 and 7), suggesting that the real-space structure of the Wannier function is unlikely to govern the observed effect.”

Comment R2.5 *It is also important that one chooses the correct unit cell – The authors here assume a unit cell that does not have the CDW reconstruction, as evidenced by the lack of backfolding in the ARPES data, however the STM topographies show a very strong CDW unit cell which would modify the QPI. If this CDW basis is chosen for their model and a continuum T-matrix calculation is performed, then one would be able to simulate the CDW bragg peaks and any scattering vectors associated with them, as well as their intensities. This change of basis would also fold the scattering vectors from the hotspot model and offer a much simpler explanation for their observation, without needing to invoke momentum dependent T-matrices. In short there are more trivial explanations for the discrepancy between the simulation and experiment that the authors have not ruled out.*

Answer to R2.5 Folding the bands by the CDW wave-vector certainly happens, but this is not enough to guarantee that the folded bands will be visible in ARPES and QPI, as one has to consider the relative spectral weight between the unfolded and the folded bands. Our point is not that the bands are not folded, but that the spectral weight of the folded bands, as obtained from ARPES, is negligible. Therefore, they are not expected to be manifested in the QPI spectrum for a featureless T -matrix.

To address this issue in a clearer way, consider our simple single-band model for the CDW, whose details are explained in the response to the first referee (Response 1.3). CDW order will create a single-particle potential with a new periodicity that will be experienced by the electrons. There is no question that, structurally, this new potential causes a big change, as the Referee points out based on our STM topography. In our model, the structural changes caused by the CDW are captured by the parameter Δ .

Now, the impact of this new potential on the QPI will depend on how the electronic spectrum is impacted by it. There is a coupling constant λ that “converts” the amplitude of the structural change Δ into the amplitude of the resulting electronic potential, $\lambda\Delta$. The main impact of the new potential is to fold the bands by the CDW wave-vector \mathbf{K}_{CDW} , see Eq. (2). But this does not imply that the folded bands have the same spectral weight as the original bands, as shown in Eq. (3). In fact, as we discussed in the derivation of Eq. (6), the spectral weight of the folded bands will be very small in the momentum region with $|\xi_{\mathbf{k}} - \xi_{\mathbf{k}+\mathbf{K}_{\text{CDW}}}| \gg 2\lambda\Delta$. For very small λ , this condition can be met in a wide range of the Brillouin zone.

The fact that the ARPES data does not see the folded bands implies that they have negligible spectral weight over essentially the entire Brillouin zone. This can be captured in our simple model by a very small λ (even though Δ itself would be large) [see Supplementary Fig. 11]. But

the key point is that the QPI is determined by the same Green’s function probed by ARPES. Therefore, in the case of a featureless T -matrix, the negligible spectral weight of the folded bands will make them also “invisible” to QPI. We would like to emphasize that, while our simple model helps us make this point sharper, it is not essential. The robust conclusion is that, **assuming a momentum-independent T -matrix**, the folded bands should be either visible in ARPES and QPI or invisible in both, since it is the same electronic Green’s function that is probed in the two cases.

In the revised manuscript, we clarify these issues to the reader and clearly separate the experimental data from our theoretical discussions. We also explicitly describe alternate explanations for the experimental observations, such as the surface structure of the Wannier functions, and explain why we do not believe they are the cause of the observed effect.

Comment R2.6 *My final criticism, is that the conclusions drawn about the need for “strong correlations” are not supported by any additional simulation. The authors propose that a momentum dependent T -matrix, caused by “strong electronic fluctuations” (eq. 1) is necessary to explain the discrepancy between simulation and experiment, but do not calculate such a quantity to show that this would address the issue. Therefore rendering it a hypothesis rather than a result. This simulation should not be that difficult, T -Matrix simulations from DFT derived models are routinely performed in the literature [see e.g R1-R4]. Without that simulation to support their claim, then the titular conclusion, “Quasiparticle interference reveals strong fluctuations in the Charge Order State of ScV_6Sn_6 ” has not been demonstrated.*

Answer to R2.6 Our main point, which is more clearly stated in the revised manuscript, is that the ARPES and QPI data are contradictory to each other if one assumes a momentum-independent T -matrix, as one would generally expect to arise from point-like impurities. Our proposal that vertex corrections of the impurity potential due to bosonic fluctuations can endow a featureless T -matrix with a momentum dependence is motivated by well-established theoretical results obtained in different contexts, Refs. [62-63]. Such a vertex-correction mechanism has been previously invoked to explain STM data in other compounds [60]. A natural candidate for fluctuations peaked at the same in-plane wave-vector as the CDW comes from the subleading CDW instabilities revealed by first-principle calculations, see Refs. [44-45].

We acknowledge that our proposal is at this point a plausible hypothesis that requires further investigations, which are beyond the scope of this work. For this reason, we have revised the manuscript to better separate the data itself (which is ultimately the basis of our paper) and the hypothesis that we invoke to explain the conflict between the ARPES and QPI data.

The Referee brings up another potential mechanism that can endow the T -matrix with a momentum dependence, namely, the internal structure of the Wannier orbitals of the local atoms. As we explained above, a reliable Wannierization of the relevant orbitals involved in the electronic structure of the 166 kagome metals remains to be determined. While we now also mention this possible explanation in the revised manuscript, we do not think this can completely address the ARPES-QPI discrepancy. First, it is unlikely that the intra-unit cell structure of the Wannier orbitals will result in a T -matrix peaked at large momentum. Second, and most importantly, we observe the same QPI pattern by tunneling into two different layers (ScSn_2 and V), whose atoms are described by different Wannier orbitals. Thus, it is not obvious to us that “many trivial mechanisms could equally explain the data.” We hope that publication of our work will motivate the groups that have expertise in DFT-based QPI modeling to employ their machinery to this problem.

Comment R2.7 *I wish to stress that the ARPES, STM and optical data is of very high quality and provides important information into this new Kagome material. Highlighting the different consequences the CDW has on the electronic structure compared to the well known AV_3Sb_5 systems. Nevertheless the manuscript presents a narrative that has not been proved, and reading this one would be mistaken for assuming that exotic phenomena is present even when many trivial mechanisms could equally explain the data. I therefore do not recommend publication unless sufficient revisions are made to the discussion and conclusions of this manuscript.*

Answer to R2.7 As discussed above, we do not find other trivial explanations that can better explain our data than the hypothesis we put forward. However, we certainly do not wish to mislead the reader into mistaking our hypothesis for fact. In the revised manuscript we have made changes to the title, abstract and text in a way that makes clear what the data are, and what our hypothesis is.

Comment R2.8 *As an aside there have also been similar ARPES and STM studies appearing recently on the ArXiv (Ref. [Lee et al. arXiv:2304.11820 (2023), Cheng et al. arXiv:2302.12227 (2023)], (although no QPI), which I'm sure the authors are aware of. It would be of interest to readers if the authors could mention whether this manuscript supports or contradicts these studies e.g. as a note in a revised edition of the manuscript.*

Answer to R2.8 We are aware of the references as mentioned by the reviewer. The main conclusion of our study and the above references suggest that the physics of the CDW in ScV_6Sn_6 is rather different from the canonical kagome metal AV_3Sb_5 . The ARPES data presented in the references mentioned by the referee are broadly consistent with our own data, and all of the existing ARPES data suggest minimal change of the overall electronic structure and the position of vHS across the CDW transition, in contrast to AV_3Sb_5 , where the large shift of the vHSs were observed.

Regarding the STM results in Ref. [arXiv:2302.12227 (2023)] and ours, there are striking differences; we have seen dispersing QPI patterns on different surfaces, whereas such features were not observed in the above reference. It is therefore difficult to make any statements of comparison between the two results. The work above also shows evidence for a partial gap in the tunneling spectra for the V-terminated layer. In contrast, we do not observe any significant gap at the Fermi energy in the spectrum. Our results are consistent with the ARPES results and the optical spectroscopy data, where no such gap has been observed at the Fermi level in the CDW phase. As per the reviewer's suggestion, we have included these references in the revised version of our paper and added some discussions of the points above. We have added a note on Page 13 of the manuscript, reading "Note: During the preparation of our manuscript, we noticed two preprint papers, dealing with the ARPES and STM studies of ScV_6Sn_6 [50, 69]. The main conclusions of these two papers, including ours suggest that the physics of the CDW in ScV_6Sn_6 is rather different from the canonical kagome metal AV_3Sb_5 . All of the ARPES data on ScV_6Sn_6 suggest a minimal change in the overall electronic structure and the position of vHS across the CDW transition. The STM results of our study and Ref. [50] show striking differences; we have seen dispersing QPI patterns at different surfaces, whereas such features were not observed in Ref. [50]. Further, the tunneling spectra on V-terminated layer, reported in Ref. [50] show an energy gap of 20 meV at the Fermi level, whereas no such gap has been observed in our data."

REVIEWER COMMENTS

Reviewer #1 (Remarks to the Author):

Kundu proposes once again that after a CDW transition has altered the crystal symmetry to a new point group, that the new electronic eigenstates of this system can elastically scatter off the CDW. Specifically, Kundu hypothesizes elastic scattering described by a T-matrix that is peaked at momentum transfers at Q of the CDW. Formally what that describes is an impurity potential with the CDW wavevector. But this concept violates the Bloch Theorem: the eigenstates of a periodic lattice that is generated by appearance of a CDW, cannot be scattered by it.

Another related concept introduced here is that elastic scattering as described by Kundu Eqn 1 occurs from "normal state" hotspots to hotspots of the actual band structure after the crystal symmetry change due to CDW. Again this cannot occur in conventional solid state physics: the "normal state" band structure does not then exist because the crystal no longer has the symmetry that generated it. It is replaced at the CDW phase transition to a new crystal symmetry by a new band structure. So elastic scattering cannot occur from or to hotspots of an electronic band structure that no longer exists.

Normally I would advise to reject a paper based on these considerations but I'm mindful of the distinguished authors of this otherwise impressive study. So my suggestion is that Kundu add a paragraph pointing out to the reader the proposed violation of Bloch Theorem and then explain clearly the reasons why this could be possible and specifically why it occurs here.

Reviewer #2 (Remarks to the Author):

It is clear that the authors have seriously considered the comments and criticisms of both Referee 1 and myself. This revised manuscript now clearly separates the interesting data and important findings from their hypothesis. Whilst I still have my reservations about whether the difference between the STM and ARPES data is due to a momentum-dependent T-matrix or not, I believe the authors have sufficiently highlighted that a discrepancy between the data exists, at least at face value, and have worded their hypothesis such that it can promote further discussion and research on this topic.

I therefore recommend publication in Nature Communications.

Response to the comments about the submitted paper: *Low-Energy Electronic Structure in the Unconventional Charge-Ordered State of ScV_6Sn_6*

We thank the referees for the time and effort they spent reviewing our paper. In the following point-by-point response, the reviewers' comments are italicized, while our responses are in regular font.

Answers to Reviewer 1

Comment R1.1 *Kundu proposes once again that after a CDW transition has altered the crystal symmetry to a new point group, that the new electronic eigenstates of this system can elastically scatter off the CDW. Specifically, Kundu hypothesizes elastic scattering described by a T -matrix that is peaked at momentum transfers at Q of the CDW. Formally what that describes is an impurity potential with the CDW wavevector. But this concept violates the Bloch Theorem: the eigenstates of a periodic lattice that is generated by appearance of a CDW, cannot be scattered by it.*

Answer to R1.1 We believe that the referee is still misunderstanding our words. We certainly agree with the referee that the Bloch states do not scatter from the atomic lattice that they live on - they are after all eigenstates. But this is not what we are proposing at all. Let us try to be more clear with a specific example, done in one dimension for brevity but straightforwardly generalizable.

Consider a defect in a solid that scatters electrons between eigenstates. To lowest order, the T matrix is simply the Fourier transform of the real space potential $V(r)$ that is causing the scattering. The commonly used potential in QPI calculations is a delta function impurity.

$$V(r) = \delta(r)$$

This potential gives a T matrix that is constant in reciprocal space, as is well known. Now consider a different potential in real space:

$$V(r) = e^{-\xi|x|} \cos(Gx)$$

Here ξ is a decay constant, and G is the reciprocal lattice vector. This potential is oscillatory and has the underlying periodicity of the lattice. Such a potential can arise very naturally in a solid - for example, a charge placed on one atom can cause a response on several atoms that are nearby, with a response that gradually decays at large distances. The T matrix corresponding to this real space potential is

$$T(k) = \frac{1}{\sqrt{2\pi}} \left(\frac{\xi}{\xi^2 + (k - G)^2} + \frac{\xi}{\xi^2 + (k + G)^2} \right)$$

As is evident, this T matrix is peaked at the reciprocal lattice vector and will cause maximal scattering between states separated by this wave vector.

The generalization to the CDW case is straightforward. Simply replacing G by G_{CDW} gives a T matrix that is peaked at the CDW wave vector, as desired. Such a potential in real space can either come from a static enhancement of the CDW in the vicinity of a defect or from dynamical CDW fluctuations near the defect. There is no contradiction with Bloch's theorem.

To make this clear to the reader, we have added a sentence on page 11 of the main text, and a supplementary section 3 with the discussion above.

Comment R1.2 *Another relates concept introduced here is that elastic scattering as described by Kundu Eqn 1 occurs from "normal state" hotspots to hotspots of the actual band structure after the crystal symmetry change due to CDW. Again this cannot occur in conventional solid state physics: the "normal state" band structure does not then exist because the crystal no longer has the symmetry that generated it. It is replaced at the CDW phase transition to a new crystal symmetry by a new band structure. So elastic scattering cannot occur from or to hotspots of an electronic*

band structure that no longer exists.

Answer to R1.2 Again, this is a misconception on the part of the referee and we would like to clarify.

In the presence of the CDW, the Green's function can be written as the sum of two pieces:

$$G_{CDW} = G_0 + G_f$$

Here G_0 is a piece that is similar to the normal state bandstructure in the limit of a weak CDW, and G_f is a piece that looks like the “folded” bandstructure, due to the CDW. We have already given explicit formulas for G_0 and G_f previously.

The QPI correction to the Green's function is

$$\delta G = G_{CDW} T G_{CDW}$$

where reciprocal space labels and integrals are omitted for clarity.

Looking at the expression for the CDW Green's function above, it is evident that we will have terms of the form

$$\delta G = G_0 T G_f$$

and it is precisely these terms that we are considering in our work. Note that the G_0 and G_f come from different Bloch states in this expression - ie, the scattering is not from the “unfolded” to the “folded” part of a single Bloch state, but rather from the “unfolded” part of one Bloch state to the “folded” part of a different Bloch state, which is always allowed.

REVIEWER COMMENTS

Reviewer #2 (Remarks to the Author):

I agree with Referee 1 that the way the "momentum dependent T-matrix" was proposed violates Bloch's theorem. This issue lies in the choice of unit cell. Referee 1 is arguing that the CDW will modify the unit cell basis, and the scattering vectors from the QPI should also be reconstructed to match these new reciprocal lattice vectors. Whereas Kundu et al. are arguing that the unit cell basis can be thought of as unchanged, and that the CDW is just a weak perturbation on top of this. Formally Referee 1 is correct, a CDW will modify the unit cell basis, no matter how weak the potential, and the k and q vector basis must be reconstructed to account for that.

The challenge comes in reconciling the ARPES data with the QPI data – which is a central point of this manuscript. The ARPES data shows only a very weak reconstruction across the CDW transition. This does not mean that the k -space Brillouin zone and reciprocal lattice vectors have not been reconstructed, they have to by definition, it just means that the spectral weight is projected almost entirely onto the original non-CDW k -vectors, which is evidence in support of Kundu et al.'s argument for a perturbative modification of the non-CDW Green's function using the undistorted unit cell basis.

However the STM measurements show a complete reconstruction of the surface due to the CDW (e.g. the topographies of Fig. 1c), which can not be explained by a weak modification. Moreover, the important QPI scattering vectors observed experimentally can equally be explained by performing a co-ordinate transformation of the k -space vectors from the original unit cell to the new CDW unit cell. In this case the reconstruction of the Brillouin zone and Reciprocal lattice vectors would be equivalent to a translation of the q -vectors by K_{cdw} (as discussed in Fig 4f) but the T-matrix would be momentum independent (thus not violating Bloch's Theorem).

Why angle resolved photoemission spectroscopy only observes a weak modification from the CDW but the local density of states experiences a strong modification is in my opinion still an open question that highlights a gap in our fundamental understanding of what it is that ARPES and QPI actually measure and how we can compare the two measurements. To address this I believe goes beyond looking at what scattering vectors are allowed in QPI (e.g. the JDOS) and requires an in-depth understanding of the relative intensities of each scattering vector, from e.g. Continuum local density of states simulations as discussed in my previous report.

Although I do not believe that the distinguished natures of the referees has anything to do with whether we should reject or accept this manuscript, I do agree that the experimental data is very high quality and of interest to a wide audience. I would still recommend for publication, but suggest that the authors revise the text to focus less on the momentum-dependent T-matrix as a new piece of physics, but instead highlight the discrepancy between the ARPES and STM measurements. If they choose to still discuss the momentum dependent T-matrix as a viable hypothesis, they should explicitly state that this formally violates Bloch's theorem, and justify why this is still a valid concept to discuss.

Response to the comments about the submitted paper: *Low-Energy Electronic Structure in the Unconventional Charge-Ordered State of ScV_6Sn_6*

We have modified our manuscript to (a) focus on the main experimental points of the paper and (b) de-emphasize any suggestion that the Bloch theorem is violated in the theoretical analysis. We have added a new paragraph to the revised manuscript and removed several sentences in response (bottom of page 9 to page 10). The thrust of the new paragraph is to emphasize the primary experimental result of the paper - that the ARPES and STM data seemingly contradict each other. This is now done without any discussion of theory, upfront in the manuscript. At the same time, we have removed sentences that discuss the T -matrix ahead of the experimental discussions. In this manner, our discussion of the T matrix is only presented after the primary experimental data are fully discussed, and it is clear to the reader as to what the experimental data say and what our hypothesis is for explaining the data.

We thank referee 2 for the time and effort they spent reviewing our paper. In the following point-by-point response, the reviewers' comments are italicized, while our responses are in regular font.

Answers to Reviewer 2

Comment R2.1 *I agree with Referee 1 that the way the “momentum dependent T-matrix” was proposed violates Bloch’s theorem. This issue lies in the choice of unit cell. Referee 1 is arguing that the CDW will modify the unit cell basis, and the scattering vectors from the QPI should also be reconstructed to match these new reciprocal lattice vectors. Whereas Kundu et al. are arguing that the unit cell basis can be thought of as unchanged, and that the CDW is just a weak perturbation on top of this. Formally Referee 1 is correct, a CDW will modify the unit cell basis, no matter how weak the potential, and the k and q vector basis must be reconstructed to account for that.*

Answer to R2.1 While we agree with the referee that the CDW reconstructs the electronic structure, resulting in a new unit cell in real and reciprocal space, we disagree with the referee’s characterization of our discussion of the QPI. We use the formal expression

$$\delta G = G_a T G_b$$

(subscripts and integrals omitted for clarity). In this expression, G_a and G_b are Green’s functions for two different Bloch states in the reconstructed bands after the CDW is formed. This is the textbook definition of the scattering process, and does not violate Bloch’s theorem. From this starting point, we highlight the fact that the scattering process observed in the STM experiment corresponds to a scattering between a “normal state” hotspot in G_a and a “folded” hotspot in G_b . Again, since G_a and G_b are two different Bloch states of the CDW bandstructure, there is absolutely no problem with this.

Comment R2.2 *The challenge comes in reconciling the ARPES data with the QPI data – which is a central point of this manuscript. The ARPES data shows only a very weak reconstruction across the CDW transition. This does not mean that the k -space Brillouin zone and reciprocal lattice vectors have not been reconstructed, they have to by definition, it just means that the spectral weight is projected almost entirely onto the original non-CDW k -vectors, which is evidence in support of Kundu et al’s argument for a perturbative modification of the non CDW Greens function using the undistorted unit cell basis.*

Answer to R2.2 We agree with the referee.

Comment R2.3 *However the STM measurements shows a complete reconstruction of the surface due to the CDW (e.g the topographies of Fig. 1c), which can not be explained by a weak modification. Moreover, the important QPI scattering vectors observed experimentally can equally be explained by performing a co-ordinate transformation of the k -space vectors from the original unit cell to the new CDW unit cell. In this case the reconstruction of the Brillouin zone and Reciprocal lattice vectors would be equivalent to a translation of the q -vectors by K_{cdw} (as discussed in Fig 4f) but the T-matrix would be momentum independent (thus not violating Bloch’s Theorem).*

Answer to R2.3 We are not sure exactly what the referee means here. If the referee means to say that the CDW reconstruction implies that QPI should see spots at the observed locations, we agree. It is the intensity of the spots that is the puzzle.

Comment R2.4 *Why angle resolved photoemission spectroscopy only observes a weak modification from the CDW but the local density of states experiences a strong modification is in my opinion still an open question that highlights a gap in our fundamental understanding of what it*

is that ARPES and QPI actually measure and how we can compare the two measurements. To address this I believe goes beyond looking at what scattering vectors are allowed in QPI (e.g the JDOS) and requires an in-depth understanding of the relative intensities of each scattering vector, from e.g Continuum local density of states simulations as discussed in my previous report.

Although I do not believe that the distinguished natures of the referees has anything to do with whether we should reject or accept this manuscript, I do agree that the experimental data is very high quality and of interest to a wide audience. I would still recommend for publication, but suggest that the authors revise the text to focus less on the momentum-dependent T -matrix as a new piece of physics, but instead highlight the discrepancy between the ARPES and STM measurements. If they choose to still discuss the momentum dependent T -matrix as a viable hypothesis, they should explicitly state that this formally violates Bloch's theorem, and justify why this is still a valid concept to discuss.

Answer to R2.4 In our modified manuscript, as discussed above, we have clearly pointed out this central discrepancy between STM and ARPES data. Our hypothesis for resolving the discrepancy has been modified to make clear that we are not suggesting a violation of Bloch's theorem, and we do acknowledge that our hypothesis is qualitative and therefore alternative explanations can certainly be entertained if they can provide a satisfactory explanation.

REVIEWERS' COMMENTS

Reviewer #2 (Remarks to the Author):

The manuscript is now written clearly to separate between the experimental study and the theoretical proposal to explain the apparent experimental discrepancy. I can again recommend for publication.